# UniQL: Unified Quantization and Low-rank Compression for Adaptive Edge LLMs

**Hung-Yueh Chiang** [1] ✉ **, Chi-Chih Chang** [2] **, Yu-Chen Lu** [3] **, Chien-Yu Lin** [4] **,**
**Kai-Chiang Wu** [3] **, Mohamed S. Abdelfattah** [2] **, Diana Marculescu** [1] ✉

[1] Chandra Family Department of Electrical and Computer Engineering,
  The University of Texas at Austin
[2] Department of Electrical and Computer Engineering, Cornell University
[3] Department of Computer Science, National Yang Ming Chiao Tung University
[4] The Paul G. Allen School of Computer Science and Engineering, University of Washington

## Abstract

Deploying large language models (LLMs) on mobile platforms faces significant challenges due to the limited memory and shared computational resources of the device. Resource availability may be an issue as it is directly impacted by on the current device workload, adding to the uncertainty of model deployment. We introduce **UniQL**, a **uni**fied *post-training* **q**uantization and **l**ow-rank compression framework, with on-device configurable pruning rates for edge LLMs. UniQL is a general framework that integrates quantization and low-rank compression for **Transformers**, **State Space Models (SSMs)**, and **hybrid models** to cater to diverse edge applications. In our proposed joint framework, we introduce an efficient *structured weight-sorting* that speeds up the computation by $20\times$, *quantization-aware* singular value decomposition (SVD) decompositions to minimize the quantization errors, *state-aware* weight sorting for SSMs, and a *fused* rotary embedding (RoPE) kernel for the pruned models. Our framework performs weight-sorting, fine-tuning, and quantization in the cloud in a **one-pass** fashion, while enabling **on-device** configurable pruning rates up to $35\%$. Our experiments show that quantized and pruned models offer a memory reduction of $4\times$–$5.7\times$ and a token throughput improvement of $2.7\times$–$3.4\times$, maintaining accuracy within 5% of the original models at $15\%$ pruning rates across Transformers (Llama3 and Qwen2.5), SSMs (Mamba2), and hybrid models (Nemotron-H and Bamba-v2). The code and quantized models will be released at: https://github.com/enyac-group/UniQL.

## 1 Introduction

Numerous emerging applications, such as question answering on VR/AR glasses, are powered by large language models (LLMs). Yet, models with parameters on the order of billions (*e.g.,* 10B) restrict the platforms and applications that can utilize them. Extensive research investigates quantization (Xiao et al., 2023; Lin et al., 2024a;b) and compression (Qinsi et al., 2025; Wang et al., 2025b; Lin et al., 2025) for LLMs to lower memory and computing needs for deployment. However, the limited and shared resources (*e.g.,* the unified memory architecture) on edge devices still pose huge challenges for model deployment. Since resources (*e.g.,* memory) are dynamically managed by the operating system, the availability of the resources highly depends on the system workload. As a result, the pre-compressed or pre-quantized language models with *fixed* model sizes may not run on a device under high workload scenarios.

Re-compressing or re-quantizing the model to fit it into available memory is not practical due to the high computational costs, *i.e.,* several hours on cloud GPUs (Lin et al., 2025; Frantar et al.,

---

✉ Corresponding authors: {hungyueh.chiang, dianam}@utexas.edu

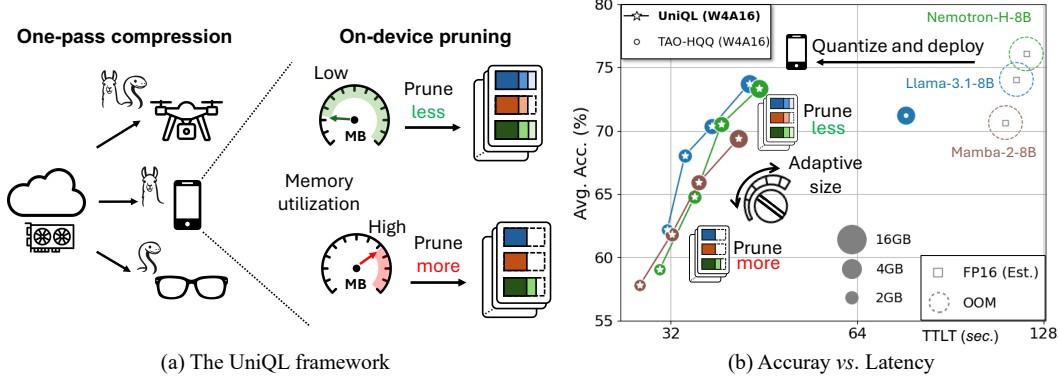

(a) The UniQL framework        (b) Accuray *vs*. Latency

Figure 1: **(Proposed framework overview.)** UniQL supports Transformers, SSMs, and hybrid models, enabling one-shot compression using a single server-class GPU. The on-device pruning of the quantized model is feasible and configurable based on the current device workload. We present actual latency on Nano 8G in relation to accuracy for different pruning rates across three distinct models on the right. Circle sizes correspond to model sizes.

2023). A solution to address this issue is storing several model replicas at different compression rates. Nonetheless, producing pre-compressed replicas of different sizes is both time- and storage-consuming. Alternatively, employing elastic training (Cai et al., 2024; 2025) to a pre-trained model enables the derivation of various sizes from the model. Yet, this approach requires availability of GPU resources and training on curated datasets to support flexible deployment for *one* specific type of model, *e.g.*, Llama-3.1-8B, limiting the applicability.

Our proposed work addresses this issue under the post-training setting when access to server-class GPUs and curated datasets is limited. As illustrated in Figure 1, our framework supports *quantization* and *structured pruning*, performing efficiently on one server GPU. Our objective is to support and design compression algorithms for various model architectures, including Transformers, State Space Models (SSMs), and hybrid models. Our pipeline is shown in Figure 2. We group the weights within the block, gather channel corrections from a calibration dataset, and apply weight-sorting algorithms. Our multi-layer perceptron (MLP) weights are decomposed without any gradient information or expensive full matrix pseudo-inverse, yielding a speedup of $20\times$ compared to prior art (Lin et al., 2025). For $W_v$ and $W_o$ in self-attention layers, we develop a quantization-aware singular value decomposition (SVD) of weights to minimize quantization errors. For SSMs and hybrids models, we find that SSM blocks are particularly sensitive to state matrices, and propose a state-aware weight-sorting strategy to mitigate this. We then apply a masked fine-tuning to the sorted model. In each fine-tuning step, a global pruning rate $P_t$ is chosen randomly, masking the least ranked channels in the layers. The refined model is then quantized in low bit-width and deployed on the edge platform. The entire process is performed *once* in the cloud. For the deployed model, we prune the models according to a specified global pruning rate, *e.g.*, $P_{35} = 35\%$, on the edge device. Our contributions are summarized as follows:

- Our study explores **a broad spectrum of models**, such as Transformers, SSMs, and hybrid, and introduces efficient pruning and quantization-friendly algorithms for these blocks.

- To the best of our knowledge, UniQL is the first post-training framework that systematically combines **quantization and structured pruning** for LLMs in a one-shot fashion.

- We develop an integrated kernel to support the pruned RoPE, conducting comprehensive profiling to demonstrate $2.7\times$–$3.4\times$ **latency speedups** for adaptive pruning on edge devices.

## 2 RELATED WORK

**Transformer compression.** Prior work has aimed to reduce the size of Transformer-based LLMs for efficient deployment by utilizing low bit-width data types (Xiao et al., 2023; Lin et al., 2024b;a; Zhao et al., 2024; Liu et al., 2025; Ashkboos et al., 2024b), minimizing storage needs and optimizing hardware for low-bit computations. Unstructured (Frantar & Alistarh, 2023; Sun et al., 2024) and

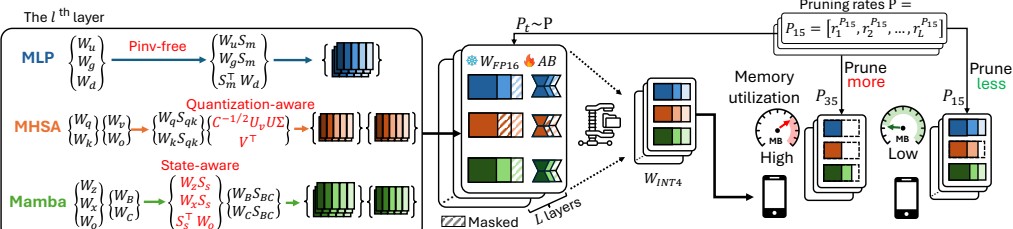

(a) Decompose and sort the channels    (b) Masked fine-tune   (c) Fuse and quantize   (d) Deploy and adaptively prune

Figure 2: **(The UniQL pipeline.)** We devise pseudo-inverse-free, quantization-aware, and state-aware matrix decomposition methods for the grouped weights to obtain sorted weights (a). During fine-tuning, we sample global pruning rates, and masked out the weight channels (b). The refined patches are fused into the weights, followed by model quantization for deployment (c). Based on the system utilization, we perform on-device adaptive pruning of the quantized model (d).

semi-structured pruning (*e.g.,* N:M sparsity) (Li et al., 2023) for reducing model size by removing specific parameters while minimizing accuracy loss. Nonetheless, deploying such methods requires specialized hardware (Taka et al., 2025; Xia et al., 2023). Structured pruning (Wang et al., 2025a; Lin et al., 2025; Ma et al., 2023; Ashkboos et al., 2024a) removes whole elements (*e.g.,* channels and heads), enabling faster inference on standard hardware but sacrificing performance. Some studies focus on one-shot compression (Genzel et al., 2025; Wang et al., 2025c), flexible bit-width quantization (Park et al., 2024), and quantization with semi-structured sparsity (Mozaffari et al., 2025). Our framework is systematically designed for quantization and on-device structured pruning.

**SSM compression.** State Space Models are memory-efficient alternatives to Transformers. Recent studies (Xu et al., 2025; Chiang et al., 2025b;a) introduce low-bit quantization techniques for SSMs. Structured (Taghibakhshi et al., 2025; Muñoz et al., 2025) and unstructured pruning (Tuo & Wang, 2025; Shihab et al., 2025) strategies have been developed for SSMs. For example, Taghibakhshi et al. (2025) eliminate the SSM heads and restore performance through knowledge distillation training. Muñoz et al. (2025) explore block-wise (*e.g.,* Mamba and Transformer blocks) and module-wise (*e.g.,* SSM and self-attention modules) structured pruning methods. Some work explores the token pruning (Zhan et al., 2024) or dimension reduction (Chi et al., 2024) for vision SSMs. Our focus is on analyzing a broader structured pruning and quantization framework for Transformers, SSMs, and hybrids, distinguishing it from previously mentioned approaches.

**Elastic training for LLMs.** Elastic training aims to enable a pre-trained LLM to dynamically adapt to varying deployment constraints such as memory, compute, and latency budgets. Flextron (Cai et al., 2024) and LLaMaFlex (Cai et al., 2025) introduce many-in-one architectures through pruning and weight sharing, allowing adaptive inference under dynamic resource constraints. Jet-Nemotron (Gu et al., 2025) further leverages post-training neural architecture search to generate compact LLM variants. These methods require GPU resources and training on curated datasets for flexible deployment tailored to a particular model type and size, *e.g.,* Llama-3.1-8B, thereby restricting their general applicability. In contrast, our work focuses on post-training on a single server GPU for the most common model architectures and supports on-device adaptation.

## 3 PROPOSED FRAMEWORK: UNIQL

### 3.1 NOTATIONS

Let $T$ represent the sequence length. $D_{\mathrm{h}}$, $D_{\mathrm{hd}}$, $D_{\mathrm{s}}$, and $D_{\mathrm{int}}$ denote the hidden, head, state, and intermediate dimensions used in Transformer and Mamba blocks. $D'$ is the post-pruning dimension. $H_s$, $H_{kv}$, and $H_m$ are the number of attention heads, key-value heads, and SSM heads, respectively. $G_s$ is the number of SSM groups. $\mathbf{X} \in \mathbb{R}^{T \times D_{\mathrm{in}}}$ and $\mathbf{W} \in \mathbb{R}^{D_{\mathrm{in}} \times D_{\mathrm{out}}}$ are the activations and weights. $\mathcal{C} = \mathbf{X}^{\top}\mathbf{X} \in \mathbb{R}^{D_{\mathrm{in}} \times D_{\mathrm{in}}}$ is the correlation matrix of $\mathbf{X}$. The matrix $\mathbf{S} \in \mathbb{R}^{D \times D}$ is associated with a group of weights for sorting their columns and rows. We denote the element-wise multiplication, broadcasted outer product, and activation function as $\odot$, $\otimes$ and $\sigma(\cdot)$, respectively. $\mathbf{U}$, $\mathbf{\Sigma}$, and $\mathbf{V}$ denote the SVD decomposition, with eigenvalues $\sigma_i$ on $\mathbf{\Sigma}$'s diagonal.

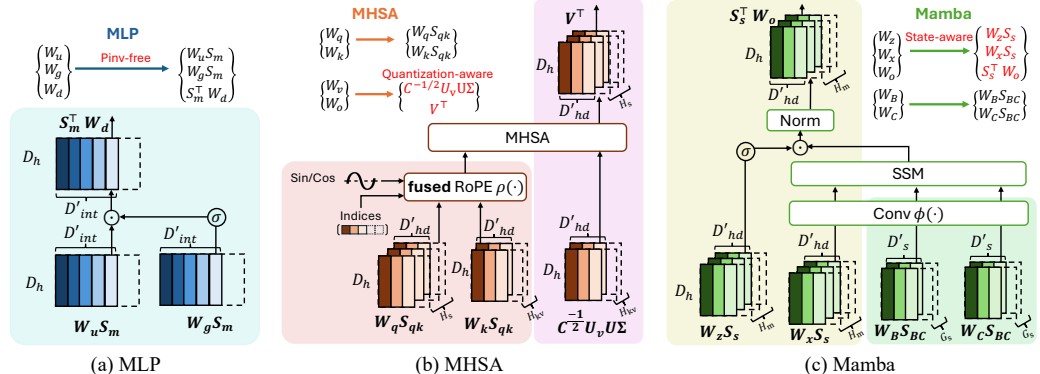

Figure 3: **(Joint weight decomposition.)** We visualize the group of sorted *weights* in MLP (a), MHSA (b), and Mamba (c) blocks. The group of weights for joint decomposition is shown in the same background color, *e.g.,* $\mathbf{W}_q$ and $\mathbf{W}_k$ in the pink background, and other groups are distinguished by different colors. We devise different types of joint compression algorithms that are efficient and quantization-aware to support on-device pruning.

## 3.2 STRUCTURED WEIGHT SORTING

Our objective is to enable adaptive on-device pruning by sorting the weights according to their importance scores, allowing the device to prune the least significant columns. Inspired by recent studies (Lin et al., 2025; Koike-Akino et al., 2025), we group the weights and conduct joint decomposition, as shown in Figure 3. We co-design the pruning algorithm alongside quantization and fused kernels for Transformers, SSMs, and hybrids.

**Multi-layer perceptron (MLP).** The MLP includes up $\mathbf{W}_u \in \mathbb{R}^{D_{\mathrm{h}} \times D_{\mathrm{int}}}$ and down projections $\mathbf{W}_d \in \mathbb{R}^{D_{\mathrm{int}} \times D_{\mathrm{h}}}$, with an optional gate projection $\mathbf{W}_g \in \mathbb{R}^{D_{\mathrm{h}} \times D_{\mathrm{int}}}$. The formulation is defined as $f_{\mathrm{MLP}}(\mathbf{X}) = (\sigma(\mathbf{X}\mathbf{W}_g) \odot \mathbf{X}\mathbf{W}_u)\mathbf{W}_d$. To derive $\mathbf{S}_m$ to sort the weight matrices in the MLP layer, we collect the intermediate activation $\mathbf{X}_{\mathrm{int}} = \sigma(\mathbf{X}\mathbf{W}_g) \odot \mathbf{X}\mathbf{W}_u$ from the calibration set, and calculate the channel correlation $\mathcal{C} = \mathbf{X}_{\mathrm{int}}^\top \mathbf{X}_{\mathrm{int}} \in \mathbb{R}^{D_{\mathrm{int}} \times D_{\mathrm{int}}}$. We average the correlation matrix over the calibration set, and compute the ridge leverage scores (McCurdy, 2018) defined by $\mathrm{diag}\left(\mathcal{C}(\mathcal{C} + \lambda I)^{-1}\right)$. We set ridge lambda $\lambda = 1$ in our experiments. We use the scores and create a column sorting matrix $\mathbf{S}_m \in \mathbb{R}^{D_{\mathrm{int}} \times D_{\mathrm{int}}}$ that reorders the output columns for $\mathbf{W}_u$ and $\mathbf{W}_g$ as $\mathbf{W}_u\mathbf{S}_m$ and $\mathbf{W}_g\mathbf{S}_m$, and the input rows of the $\mathbf{W}_d$ as $\mathbf{S}_m^\top \mathbf{W}_d$, as shown in Figure 3 (a).

Table 1: **(Pseudo-inverse.)** Pseudo-inverse latency for FP64 matrices on A6000 (in minutes).

| Matrix Size | Lat. (min.) |
|---|---|
| [1024, 1024] | 0.02 |
| [4096, 4096] | 0.57 |
| [8192, 8192] | 4.24 |
| [14336, 14336] | 20.58 |

Our approach does not rely on time-consuming pseudo-inverse to sort the MLP weight matrices. Although the pseudo-inverse (*i.e.,* Moore-Penrose inverse (Penrose, 1955)) provides a theoretical bound in pruning errors (Lin et al., 2025), it exhibits three major drawbacks: **(1)** Pseudo-inverse has a complexity of $O(n^3)$ for a $n$-size squared matrix. This is particularly time-consuming when computing the pseudo-inverse of correlation matrices in MLP layers because $D_{\mathrm{int}}$ is a large number in most LLM designs, *e.g.,*, Llama-3-8B $D_{\mathrm{int}} = 14336$. **(2)** Pseudo-inverse computation requires a high-precision FP64 to maintain numerical stability (Lin et al., 2025), which demands substantial memory usage for full-precision weights. **(3)** Matrix inverse breaks the equivalence of pruned weights, resulting in $(\mathbf{W}')^\dagger \neq (\mathbf{W}^\dagger)'$, requiring recomputation for *different* pruning rates. Here, $\mathbf{W} \in \mathbb{R}^{D \times D}$. $\mathbf{W}'$ is a submatrix of $\mathbf{W}$, where $\mathbf{W}' \in \mathbb{R}^{D \times D'}$ with $D > D'$. $\mathbf{W}^\dagger$ represents the inverse matrix. We show the latency of pseudo-inverse computation for FP64 matrices on A6000 in Table 1.

**Multi-head self-attention (MHSA).** For simplicity, we set the attention heads $H_s$ and the key-value heads $H_{kv}$ as equivalent. The formulation of $i^{\mathrm{th}}$ head within the MHSA is provided as fol-

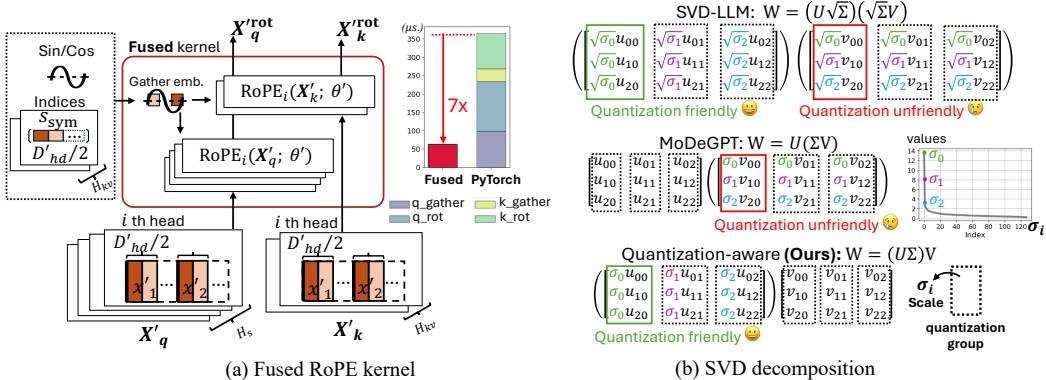

(a) Fused RoPE kernel    (b) SVD decomposition

Figure 4: **(The fused kernel and SVD decomposition.)** In the left illustration, gathering and slicing rotary positional embeddings by the index vector for $Q$ and $K$ are fused in one kernel to reduce memory access. The embeddings for the pruned head dimension $D'_{\text{hd}}$ are gathered from the index array $\mathbf{S}_{sym}$ in the fused kernel. On the right, we combine the diagonal matrix $\Sigma$ with $\mathbf{U}$ as the group shares a quantization scaling factor to reduce the quantization errors.

lows: $f_{\text{MHSA}}(\mathbf{X}, i) = \text{Softmax}\left(\rho\left(\mathbf{X}\mathbf{W}_q^i\right)\rho\left(\mathbf{X}\mathbf{W}_k^i\right)^\top\right)\left(\mathbf{X}\mathbf{W}_v^i\mathbf{W}_o^i\right)$, where $\rho(\cdot)$ denotes RoPE (Su et al., 2021). The weights in MHSA are divided into two groups: $\{\mathbf{W}_q^i, \mathbf{W}_k^i\}$ and $\{\mathbf{W}_v^i, \mathbf{W}_o^i\}$.

For the $\mathbf{W}_q^i$ and $\mathbf{W}_k^i$, we obtain activations $\mathbf{X}_q^i = \rho\left(\mathbf{X}\mathbf{W}_q^i\right)$ and $\mathbf{X}_k^i = \rho\left(\mathbf{X}\mathbf{W}_k^i\right)$, and compute the channel correlation $\mathcal{C}_q^i = \mathbf{X}_q^{i\top}\mathbf{X}_q^i$ and $\mathcal{C}_k^i = \mathbf{X}_k^{i\top}\mathbf{X}_k^i$, where $\mathcal{C}_q^i$ and $\mathcal{C}_k^i \in \mathbb{R}^{D_{\text{hd}} \times D_{\text{hd}}}$. The sorting scores $s \in \mathbb{R}^{D_{\text{hd}}}$ are calculated as $s = \|\mathcal{C}_q^{i\,1/2}\| \odot \|\mathcal{C}_k^{i\,1/2}\|$, and averaged over the calibration sample. Since the embedding positions are broken by our structured sorting, we have to gather the corresponding indices for the rotary positional embeddings, *i.e.,* sin and cos. Since RoPE is expressed as $\text{RoPE}(\mathbf{X}; \theta) = \cos(\theta) \odot \mathbf{X} + \sin(\theta) \odot \text{R}(\mathbf{X})$, where $\text{R}(\mathbf{X})$ rotates by splitting $\mathbf{X}$ into two components along the last axis: $\mathbf{X} = [\mathbf{x}_1, \mathbf{x}_2]$ such that $\text{R}(\mathbf{X}) = [-\mathbf{x}_2, \mathbf{x}_1]$, we apply a symmetric sorting $s_{\text{sym}} = s_1 + s_2$, where $\{s_{\text{sym}}, s_1, s_2\} \in \mathbb{R}^{D_{\text{hd}}/2}$ and $s = [s_1, s_2]$. As such, we construct the final sorting matrix $\mathbf{S}_{qk} \in \mathbb{R}^{D_{\text{hd}} \times D_{\text{hd}}}$ with $[s_{\text{sym}}, s_{\text{sym}}]$ that sorts the output columns as $\mathbf{W}_q\mathbf{S}_{qk}$ and $\mathbf{W}_k\mathbf{S}_{qk}$. Symmetric sorting is hardware-efficient since we only need to store and load half of the index vector into our fused RoPE kernel, as illustrated in Figure 4 (a).

For the $\mathbf{W}_v^i$ and $\mathbf{W}_o^i$, we perform an activation-scaled SVD decomposition (Wang et al., 2025a; Yuan et al., 2023). With the input correlation matrix $\mathcal{C} = \mathbf{X}^\top\mathbf{X}$, we follow Lin et al. (2025) to perform joint decomposition by two consecutive SVD operations: $\mathcal{C}^{\frac{1}{2}}\mathbf{W}_v^i\mathbf{W}_o^i = \text{SVD}(\mathcal{C}^{\frac{1}{2}}\mathbf{W}_v^i)\mathbf{W}_o^i = \mathbf{U}_v\mathbf{\Sigma}_v\mathbf{V}^\top_v\mathbf{W}_o^i = \mathbf{U}_v\text{SVD}(\mathbf{\Sigma}_v\mathbf{V}^\top_v\mathbf{W}_o^i) = \mathbf{U}_v\mathbf{U}\mathbf{\Sigma}\mathbf{V}^\top$. The SVD decomposition ranks the eigenvectors by eigenvalues. To reduce quantization errors, we fuse the diagonal matrix $\mathbf{\Sigma}$ into $\mathbf{W}_v^i$, unlike prior art (Wang et al., 2025b; Lin et al., 2025). We present the final sorted weights with quantization-aware SVD decomposition as $\mathbf{W}_v^i = \mathcal{C}^{\frac{-1}{2}}\mathbf{U}_v\mathbf{U}\mathbf{\Sigma}$ and $\mathbf{W}_o^i = \mathbf{V}^\top$, as shown in Figure 3 (b). Low-bit quantization (*i.e.,* 3- or 4-bit) is sensitive to the distribution within the quantization group. We deconstruct the weight matrix $\mathbf{W} = \mathbf{U}\mathbf{\Sigma}\mathbf{V}$ by merging the *long-tailed* eigenvalues $\mathbf{\Sigma}$ with $\mathbf{U}$, such that $\mathbf{W} = (\mathbf{U}\mathbf{\Sigma})\mathbf{V}$, where each column of $\mathbf{U}$ is scaled by its eigenvalue $\sigma_i$. Thus, $\sigma_i$ acts as the scaling factor for the group *without* distorting the distributions, as depicted in Figure 4 (b).

We show the details of the proposed algorithms in Algorithm 2 and 3. This joint weight decomposition also supports Grouped-Query Attention (GQA) (Ainslie et al., 2023), as shown in Figure 3 (b) and Algorithm 4 and 5 in Appendix A.

**Mamba.** The Mamba block encompasses five primary weight matrices. For simplicity, we decompose the entire computation and express the SSM's $i^{\text{th}}$ head and $g^{\text{th}}$ group as $f_{\text{Mamba}}(\mathbf{X}, i, g) = \text{Norm}\left(\sigma(\mathbf{X}\mathbf{W}_z^i) \odot \text{SSM}\left(\Delta\mathbf{A}, \phi(\mathbf{X}\mathbf{W}_C^g), \Delta\phi(\mathbf{X}\mathbf{W}_B^g), \phi(\mathbf{X}\mathbf{W}_x^i)\right)\right)\mathbf{W}_o^i$, where $\{\mathbf{W}_x^i, \mathbf{W}_z^i\} \in \mathbb{R}^{D_{\text{h}} \times D_{\text{hd}}}$, and $\{\mathbf{W}_B^g, \mathbf{W}_C^g\} \in \mathbb{R}^{D_{\text{h}} \times D_{\text{s}}}$, and the output weight $\mathbf{W}_x^i \in \mathbb{R}^{D_{\text{hd}} \times D_{\text{h}}}$. $\text{Norm}(\cdot)$ and $\phi(\cdot)$ denote normalization and 1D causal convolution layer fused with an activation, respectively. The

SSM($\cdot$) function performs the linear recurrence computations $h_t = \Delta_t A_t h_{t-1} + \Delta_t B_t x_t, \quad y_t = C_t h_t$ for the $i^{\text{th}}$ head and $g^{\text{th}}$ group at each time step $t$ with a parameterized step size $\Delta$ (Dao & Gu, 2024). $\Delta\mathbf{A}$ and $\Delta\mathbf{B}^g = \Delta\phi(\mathbf{XW}_B^g)$ are the matrix forms of $\Delta_t A_t$ and $\Delta_t B_t$. $\mathcal{H}$ is the matrix form of the SSM state $h_t$. To perform the joint weight decomposition, we first break the computation of a Mamba block into two sub-formulations: (1) SSM input mask $\mathcal{M}$: $f_{\mathcal{M}}(\mathbf{X}, g) = \mathbf{C}^g(\Delta\mathbf{B}^g)^\top = \phi(\mathbf{XW}_C^g)(\Delta\phi(\mathbf{XW}_B^g))^\top$ and (2) SSM state $\mathcal{H}$: $f_{\mathcal{H}}(\mathbf{X}, g, i, h_0) = \Delta\mathbf{A}\mathcal{H}(h_0) + \Delta\mathbf{B}^g\mathbf{X}_\phi^i$ with an initial state $h_0$. We denote $\mathbf{X}_\phi^i = \phi(\mathbf{XW}_x^i)$ as the output of the causal convolution activation.

We first focus on sorting the weights $\{\mathbf{W}_B^g, \mathbf{W}_C^g\}$ that compute the SSM input mask $\mathcal{M}$. For the SSM group $g$ with $H_m^g = \frac{H_m}{G_s}$ SSM heads in the group, we obtain the activations $\mathbf{B}^g = \phi(\mathbf{XW}_B^g)$ and $\mathbf{C}^g = \phi(\mathbf{XW}_C^g)$, where $\mathbf{B}^g$ and $\mathbf{C}^g \in \mathbb{R}^{T \times D_s}$. Unlike self-attention, $\mathbf{B}^g$ is discretized using the input-dependent variable $\Delta^g \in \mathbb{R}^{T \times H_m^g}$ by a broadcasted outer product $(\Delta\mathbf{B})^g = \Delta^g \otimes \mathbf{B}^g \in \mathbb{R}^{H_m^g \times T \times D_s}$. As a result, we compute the channel correlation $\Delta\mathcal{C}_B^g = (\Delta\mathbf{B})^{g\top}(\Delta\mathbf{B})^g$ and $\mathcal{C}_C^g = \mathbf{C}^{g\top}\mathbf{C}^g$, where $\Delta\mathcal{C}_B^g \in \mathbb{R}^{H_m^g \times D_s \times D_s}$ and $\mathcal{C}_C^g \in \mathbb{R}^{D_s \times D_s}$. The sorting scores $s \in \mathbb{R}^{D_s}$ are calculated as $s = \sum_{\tau=0}^{H_m^g} \|(\Delta\mathcal{C}_B^g)_\tau^{1/2}\| \odot \|\mathcal{C}_C^{g,1/2}\|$, and averaged over the calibration samples. We use $s$ to construct the sorting matrix $\mathbf{S}_{BC} \in \mathbb{R}^{D_s \times D_s}$ that sorts the output columns of $\mathbf{W}_B^g$ and $\mathbf{W}_C^g$ as $\mathbf{W}_B^g\mathbf{S}_{BC}$ and $\mathbf{W}_C^g\mathbf{S}_{BC}$, as shown in Figure 3 (c).

We propose a *state-aware* method to compress the other group of weights $\{\mathbf{W}_z^i, \mathbf{W}_x^i, \mathbf{W}_o^i\}$ by collecting the correlations from the SSM states, $\mathcal{H}^i \in \mathbb{R}^{(T \times D_s) \times D_{\text{hd}}}$, such that $\mathcal{C}_{\mathcal{H}}^g = \mathcal{H}^{i\top}\mathcal{H}^i$, $\mathcal{C}_C^g \in \mathbb{R}^{D_{\text{hd}} \times D_{\text{hd}}}$ and averaging over the calibration samples. The ridge leverage score $\text{diag}\left(\mathcal{C}_{\mathcal{H}}(\mathcal{C}_{\mathcal{H}} + \lambda I)^{-1}\right)$ is computed as the MLP layer, where we set ridge lambda $\lambda = 1$. We use the scores and design a column sorting matrix $\mathbf{S}_s \in \mathbb{R}^{D_{\text{hd}} \times D_{\text{hd}}}$ that organizes the output columns for $\mathbf{W}_z^i$ and $\mathbf{W}_x^i$ as $\mathbf{W}_z^i\mathbf{S}_s$ and $\mathbf{W}_x^i\mathbf{S}_s$. The input rows for $\mathbf{W}_o^i$ are sorted accordingly by $\mathbf{S}_s^\top\mathbf{W}_o^i$. Figure 3 (c) illustrates these sorted weights.

### 3.3 Masked LoRA Fine-tuning

We conduct a LoRA-based (Hu et al., 2022) recovery fine-tuning (FT) on the sorted model. Unlike previous work (Wang et al., 2025b;a) that fine-tunes the pruned model, we fine-tune the *un-pruned* sorted model in *one shot*, as shown in Figure 2. We derive the layer-wise pruning rates $r_l$ using Block Influence (BI) scores (Lin et al., 2025; Men et al., 2024) for all pre-determined global pruning rates $P = [P_{15}, P_{20}, ...]$, such as $P_{15} = [r_1^{P_{15}}, r_2^{P_{15}}, ..., r_L^{P_{15}}]$. The BI score is defined as $s = 1 - \mathbb{E}\frac{\mathbf{x}_l^\top\mathbf{y}_l}{\|\mathbf{x}_l\|_2\|\mathbf{y}_l\|_2}$, where the $x_l$ and $y_l$ represent the input and output of the $l^{\text{th}}$ layer, respectively. During fine-tuning, we randomly draw a pruning rate $P_t \sim P$ at time step $t$ to mask out the pruned channels. We follow the prior work (Wang et al., 2025b) to perform instruction tuning on the Alpaca dataset (Taori et al., 2023) for five epochs. The entire fine-tuning process is conducted on a single cloud GPU. Our sorted model provides configurable pruning rates on the device after *one-shot* masked fine-tuning. We note that our fine-tuning inherently supports downstream tasks for any application, such as summarization and question-answering datasets.

### 3.4 Quantization and On-device Adaptive Pruning

We quantize the fine-tuned full model to minimize the on-device storage needs. We employ group-wise uniform symmetric quantization, the most commonly supported method by hardware, to convert floating-point values into $N$-bit discrete form. The quantization function for a group of weight $\mathbf{W}_{(i,g)}$ in column $i$ is defined as $\overline{\mathbf{W}}_{(i,g)} = \text{Clamp}\left(\left\lfloor\frac{\mathbf{W}_{(i,g)}}{s}\right\rceil, -2^{N-1}, 2^{N-1} - 1\right)$, where $s = \text{Max}(|\mathbf{W}_{(i,g)}|)/(2^{N-1} - 1)$ is the scaling factor (*i.e.,* quantization step). For the quantization-aware SVD decomposition, we fuse the eigenvalues of the $i^{\text{th}}$ column to the group of weight factor such that $\sigma_i\mathbf{W}_{(i,g)}$. We set $N = 4$ and group size 128 and adapt GPTQ (Frantar et al., 2023) to quantize our models. We fuse Hadamard matrices into the weights and apply 4-bit quantization to the embedding and output layers. The parameters of normalization layers are fused to the weights before applying quantization. After deploying the quantized model, we prune the channels on the device by reducing the intermediate dimension $D_{\text{int}}$ in the MLP layer, head dimension $D_{\text{hd}}$ in the MHSA layer, and both the state dimension $D_s$ and head dimension $D_{\text{hd}}$ in the Mamba layer. We

Table 2: **(Compared with structured pruning.)** We compare all models in FP16. UniQL enables all compression rates in single pass. The symbols $\diamond$, $\dagger$, and $\ddagger$ denote Transformers, Mamba-Transformer hybrids, and Mamba models, respectively.

| Prun.% | Prun. Method | +FT | Llama-2 7B$^\diamond$ | Llama-3.1 8B$^\diamond$ | Qwen-2.5 7B$^\diamond$ | Bamba-v2 9B$^\dagger$ | Nemotron-H 8B$^\dagger$ | Mamba2 8B$^\ddagger$ |
|---|---|---|---|---|---|---|---|---|
| 0% | – | - | 68.8% | 74.0% | 72.4% | 74.6% | 76.0% | 70.6% |
| 15% | MoDeGPT | - | 66.2% | **72.4%** | 52.1% | - | - | - |
|  | SVD-LLM | - | 56.3% | 56.7% | 62.6% | - | - | - |
|  | **UniQL (Ours)** | - | **66.7%** | 70.5% | **69.1%** | **70.9%** | **68.9%** | **65.6%** |
|  | SVD-LLM | ✓ | 64.7% | 64.5% | 69.5% | - | - | - |
|  | **UniQL (Ours)** | ✓ | **67.2%** | **71.9%** | **70.0%** | **72.9%** | **73.0%** | **66.4%** |
| 25% | MoDeGPT | - | 63.4% | 64.9% | 40.8% | - | - | - |
|  | SVD-LLM | - | 50.8% | 45.8% | 53.2% | - | - | - |
|  | **UniQL (Ours)** | - | **63.7%** | **67.0%** | **62.1%** | **66.4%** | **60.6%** | **59.8%** |
|  | SVD-LLM | ✓ | 62.4% | 59.5% | **66.8%** | - | - | - |
|  | **UniQL (Ours)** | ✓ | **64.9%** | **69.6%** | 65.8% | **69.7%** | **67.3%** | **62.7%** |

Table 3: **(Compared with PTQ.)** We benchmark all *weight-only* PTQ methods *without* fine-tuning and pruning. $^*$ represents the FP16 embeddings and output layers as per the official implementation. $^\perp$ denotes the GPTQ (Frantar et al., 2023) implemented on all models as an additional baseline.

| PTQ Method | W-bit | Llama-2 7B$^\diamond$ | Llama-3.1 8B$^\diamond$ | Qwen-2.5 7B$^\diamond$ | Bamba-v2 9B$^\dagger$ | Nemotron-H 8B$^\dagger$ | Mamba2 8B$^\ddagger$ |
|---|---|---|---|---|---|---|---|
| FP16 | 16 | 68.8% | 74.0% | 72.4% | 74.6% | 76.0% | 70.6% |
| TRT-AWQ | 4$^*$ | 68.1% | 71.9% | 70.3% | - | - | - |
| TAO-HQQ | 4$^*$ | **68.4%** | 72.4% | **72.1%** | - | - | - |
| **UniQL (Ours)** | 4$^*$ | 68.2% | **72.9%** | 72.0% | **74.8%** | **74.9%** | **69.3%** |
| GPTQ$^\perp$ | 4 | **67.9%** | 71.3% | 70.0% | 73.6% | **74.9%** | 68.1% |
| **UniQL (Ours)** | 4 | 67.8% | **72.3%** | **71.0%** | **73.8%** | 74.8% | **69.3%** |

keep the hidden state dimension $D_h$ the same across all pruned models. For `INT4` weights, we unpack them online, prune channels, and repackage them into `INT32` for the 4-bit kernel.

# 4 EXPERIMENTAL RESULTS

## 4.1 SETUPS

**Models and setups.** We experiment with Transformers Llama-2-7B (Touvron et al., 2023), Llama-3.1-8B (Meta, 2024), Qwen-2.5-7B (Hui et al., 2024), hybrid models Nemotron-H-8B (Blakeman et al., 2025), Bamba-9B-v2 (IBM, 2025), and the SSM model Mamba-2-8B (Dao & Gu, 2024). $\diamond$, $\dagger$, and $\ddagger$ denote Transformers, hybrid and SSMs, respectively. FT, PTQ, and W-bit stand for fine-tune, post-training quantization, and the bit-width of weights. Prun. and R.size represent the pruning rate in percentage (%) and the reduction of model size ($\times$), respectively.

**Structured pruning baselines.** We compare UniQL to cutting-edge model compression methods, MoDeGPT (Lin et al., 2025) and SVD-LLM (Wang et al., 2025b). As MoDeGPT is not publicly available, we duplicate their method based on the paper and achieve similar accuracy to what was reported. We adapt SVD-LLM official implementation to experiment with Llama-2-7B, Llama-3.1-8B, and Qwen-2.5-7B for compression and quantize models. For MoDeGPT and SVD-LLM, we adhere to the hyper-parameters outlined in the papers.

**Post-training quantization baselines.** We adopt AWQ (Lin et al., 2024a) in TensorRT-MO$^*$ (TRT-AWQ) (NVIDIA, 2024; 2023) and HQQ (Badri & Shaji, 2023) in TorchAO$^*$ (torchao, 2024) (TAO-HQQ) as our quantization baselines, both W4A16 libraries ready for PTQ. We evaluate UniQL in terms of model size, average accuracy on downstream tasks, and latency on A6000 and Nano 8G against TRT-AWQ and TAO-HQQ. In our experiments, we quantize the embedding and output (*i.e.,* `lm_head`) layers to 4 bits, cutting memory usage in contrast to TRT-AWQ$^*$ and TAO-

Table 4: **(One-pass adaptive pruning.)** We evaluate UniQL in **one run** across pruning rates. R.size stands for reduction of model size ($\times$). UniQL enables all pruning rates in single pass. The symbols $\diamond$, $\dagger$, and $\ddagger$ denote Transformers, Mamba-Transformer hybrids, and Mamba models, respectively. $*$ represents the FP16 embeddings and output layers as per the official implementation. We apply GPTQ (Frantar et al., 2023) on MoDeGPT (Lin et al., 2025) denoted as $\flat$.

| Method | One pass | +FT | W bit | Prun. % | R.size ($\times$) | Llama-2 7B$^\diamond$ | Llama-3.1 8B$^\diamond$ | Qwen-2.5 7B$^\diamond$ | Bamba-v2 9B$^\dagger$ | Nemotron-H 8B$^\dagger$ | Mamba2 8B$^\ddagger$ |
|---|---|---|---|---|---|---|---|---|---|---|---|
| FP16 | - | - | 16 | 0% | 0$\times$ | 68.8% | 74.0% | 72.4% | 74.6% | 76.0% | 70.6% |
| MoDeGPT$^\flat$ | $\times$ | $\checkmark$ | 4 | 15% | 4.7$\times$ | 63.7% | 64.2% | 52.2% | - | - | - |
|  | $\times$ | $\checkmark$ | 4 | 25% | 4.7$\times$ | 60.3% | 59.3% | 48.4% | - | - | - |
| SVD-LLM | $\times$ | $\checkmark$ | 4* | 15% | 4.7$\times$ | 63.2% | 60.6% | 66.8% | - | - | - |
|  | $\times$ | $\checkmark$ | 4* | 25% | 4.7$\times$ | 59.1% | 54.2% | 64.6% | - | - | - |
| **UniQL (Ours)** | $\checkmark$ | $\checkmark$ | 4 | 0% | 4$\times$ | 67.6% | 73.6% | 72.4% | 75.1% | 73.3% | 69.3% |
|  |  |  |  | 15% | 4.7$\times$ | 65.6% | 71.4% | 68.1% | 70.3% | 70.5% | 65.8% |
|  |  |  |  | 25% | 5.3$\times$ | 63.5% | 67.7% | 64.0% | 67.4% | 64.7% | 61.8% |
|  |  |  |  | 35% | 6.1$\times$ | 61.0% | 62.7% | 58.1% | 62.7% | 59.0% | 57.7% |

HQQ* which use FP16. We also adapt GPTQ (Frantar et al., 2023) for all models and also use 4-bit quantization on the embedding and output layers as an additional baseline.

**Datasets and evaluations.** We evaluate UniQL on five zero-shot tasks with a batch size of 16, including HellaSwag (Zellers et al., 2019), PIQA (Bisk et al., 2020), ARC (Clark et al., 2018), and WinoGrande (Sakaguchi et al., 2020) using LM-EVAL (Gao et al., 2023). The average of Wino-Grande, PIQA, and ARC-easy (accuracy), and HellaSwag and ARC-challenge (length-normalized accuracy) is reported for experiments. More evaluations are placed in Appendix B.

**Implementations and environments.** Our kernels are adapted from the 4-bit kernels (Frantar et al., 2024) and RoPE kernels (Hsu et al., 2025). All computations are in BF16, except for correlation matrices for structured sorting and Cholesky decomposition for GPTQ are calculated in FP32. Detailed parameters are placed at Appendix F. The weight-sorting, masked fine-tuning, and quantization are computed on an A6000 GPU with 48GB memory in one-shot for enabling adaptive pruning on the device. We profile the latency on A6000 and Orin Nano 8GB, as our experimental cloud and edge platforms. We report the average latency of twenty profiles after five warmup runs.

## 4.2 Zero-shot Downstream Tasks

**Comparison with structured pruning.** Table 2 compares structured pruning baselines against UniQL. Without fine-tuning, UniQL outperforms both MoDeGPT and SVD-LLM in most cases, achieving strong results such as 66.7% on Llama-2-7B and 69.1% on Qwen-2.5-7B. With fine-tuning, UniQL further boosts performance, reaching 67.2% on Llama-2-7B and 70.0% on Nemotron-H-8B, surpassing SVD-LLM consistently. MoDeGPT suffers from the ill-conditioned correlation matrices $\mathcal{C} \in \mathbb{R}^{D_{int} \times D_{int}}$ and numerical instability when $D_{int} >> D_h$ with limited calibration samples, resulting in large accuracy drops in Qwen-2.5-7B $\{D_{int}, D_h\} = \{18944, 3584\}$. SVD-LLM truncates numbers of the eigenvalues in the decomposed weight matrices according to the desired compression rates, requiring fine-tuning to recover the performance. These results highlight UniQL's effectiveness in preserving task performance while enabling efficient model compression.

**Comparison with post-training quantization.** Table 6 presents the comparison of *weight-only* post-training quantization (PTQ) methods. Across all models, UniQL demonstrates highly competitive performance, matching or surpassing existing PTQ methods in several settings. For example, UniQL achieves 72.9% on Llama-3.1-8B with 4-bit layers and FP16 embedding/output layers. Notably, while TAO-HQQ slightly edges out UniQL on Llama-2-7B and Qwen-2.5-7B, UniQL is more general to different architectures, providing adaptive pruning features on-device. UniQL surpasses or equals GPTQ, the baseline we modify for all models.

**One-pass adaptive pruning.** Table 4 evaluates the one-pass adaptive pruning under 4-bit quantization with fine-tuning of UniQL. We compare UniQL against SVD-LLM baselines, which follow

---

$^*$The embedding and output layers use FP16 according to the official implementation.

a similar compression process but support only a single compression rate per run. Without any pruning, UniQL achieves competitive results, for example, 67.6% on Llama-2-7B and 73.6% on Llama-3.1-8B, while reducing the model size by 4 ×. As pruning ratios increase, UniQL maintains graceful degradation in accuracy; at 15% pruning, it still achieves 71.4% on Llama-3.1-8B and 70.5% on Nemotron-H-8B, outperforming SVD-LLM across all comparable settings. At higher compression (*e.g.,* 35% pruning), UniQL still delivers reasonable performance, such as 62.7% on Llama-3.1-8B and 57.7% on Mamba2-8B. These results demonstrate UniQL's strong adaptability, generalizing to a wide range of architectures.

## 4.3 COMPRESSION TIME

In Table 5, we compare the compression time of UniQL against MoDeGPT (Lin et al., 2025), noted for being *training-free*, and SVD-LLM (Wang et al., 2025b), which involves *fine-tuning*, both state-of-the-art algorithms for Transformer compression. Our matrix decomposition is $22\times$ (0h19m *vs.* 7h03m) faster than MoDeGPT and $1.8\times$ (0h19m *vs.* 0h35m) faster than SVD-LLM, as UniQL avoids pseudo-inverse and SVD decomposition for large MLP weight matrices. With masked fine-tuning (FT), UniQL remains quicker (6h59m) than both MoDeGPT (7h03m) and SVD-LLM (15h57m). MoDeGPT suffers from the high computation cost of performing pseudo-inverse on large weight matrices. SVD-LLM splits weights into two successive layers, $\mathbf{U}$ and $\mathbf{V}$, and carries out independent fine-tuning for each, leading to a longer fine-tuning duration. Lastly, post-training quantization (PTQ) takes an extra forty minutes. Our compression algorithm is *one-time* $O(1)$ with respect to the number of compression rates, compared to $O(n)$ for MoDeGPT and SVD-LLM.

Table 5: **(Compression time.)** The time is reported on an A6000 GPU. UniQL supports all compression rates in *one shot*.

| Method | +FT | +PTQ | Llama-3.1 8B$^\diamond$ | Mamba2 8B$^\ddagger$ |
|---|---|---|---|---|
| MoDeGPT | - | - | 7h03m | - |
| SVD-LLM | - | - | 0h35m | - |
| | ✓ | - | 16h25m | - |
| | ✓ | ✓ | 16h46m | - |
| **UniQL (Ours)** | - | - | 0h19m | 0h16m |
| | ✓ | - | 6h59m | 7h18m |
| | ✓ | ✓ | 7h43m | 7h50m |

Table 6: **(Model size.)** The model size is reported in GB. Our 4-bit embedding/output layers yield smaller size than TRT-AWQ and TAO-HQQ.

| Method | W-bit | Prun. p% | Llama-3.1 8B$^\diamond$ | Qwen-2.5 7B$^\diamond$ | Nemotron 8B$^\dagger$ |
|---|---|---|---|---|---|
| FP16 | 16 | 0% | 16.0 GB | 15.2 GB | 16.2 GB |
| MoDeGPT | 16 | 15% | 13.9 GB | 13.2 GB | – |
| SVD-LLM | 16 | 15% | 14.1 GB | 13.3 GB | – |
| TRT-AWQ | 4$^*$ | 0% | 5.8 GB | 5.6 GB | – |
| TAO-HQQ | 4$^*$ | 0% | 5.7 GB | 6.0 GB | – |
| **UniQL (Ours)** | 4 | 0% | 4.1 GB | 3.9 GB | 4.1 GB |
| | | 35% | 2.8 GB | 2.7 GB | 2.9 GB |

## 4.4 MODEL SIZE AND LATENCY PROFILING

We evaluate the model size and latency of UniQL and compare them with AWQ (Lin et al., 2024a) from TensorRT-MO$^*$ (NVIDIA, 2024; 2023) (TRT-AWQ) and HQQ (Badri & Shaji, 2023) in TorchAO$^*$ (torchao, 2024) (TAO-HQQ). Both libraries are weight-only (*i.e.,* W4A16) quantization frameworks in production. We show the model size in Table 6. UniQL quantizes models in head-to-toe fashion, *i.e.,* from embeddings, backbone layers, and output heads all at 4-bit, resulting in smaller model size (4.1GB *vs.* 5.7GB) compared to TRT-AWQ and TAO-HQQ with minimal accuracy drops. Our model enables all compression rates and on-device structured pruning, providing an elastic $3.9\times - 5.7\times$ memory reductions. We profile the time-per-output-token (TPOT, *i.e.,* generation) and time-to-last-token (TTLT, *i.e.,* prefilling and generation) on A6000 and Nano 8G, and show $2.7\times - 3.4\times$ throughput improvements in generation, outperforming TRT-AWQ and TAO-HQQ, as shown in Table 7 and Table 8. On the Nano 8G, our model is $1.7\times$ faster than TAO-HQQ in TPOT. By pruning 35% of weights in our 4-bit model, our models generate $2.1\times$ faster than TAO-HQQ.

## 5 ABLATION STUDY

We conduct an ablation study to demonstrate the effectiveness of each component of our framework. Additional ablation studies can be found at Appendix C.

---

$^*$The embedding and output layers use FP16 according to the official implementation.

Table 7: **(Latency profiling on an A6000.)** Reported TPOT and TTLT (1k+1k) are in *ms*.

| Method | W-bit | Prun. p% | Llama-3.1-8B$^\diamond$ | | Nemotron-H-8B$^\dagger$ | |
|---|---|---|---|---|---|---|
| | | | TPOT | TTLT | TPOT | TTLT |
| FP16 | 16 | 0% | 25.0 | 26653.8 | 24.4 | 25889.4 |
| TRT-AWQ | 4* | 0% | 11.4 | 10130.4 | - | - |
| TAO-HQQ | 4* | 0% | 10.2 | 11639.5 | - | - |
| **UniQL (Ours)** | 4 | 0% | 9.0 | 9944.6 | 8.2 | 9095.7 |
| | | 35% | 7.3 | 8105.4 | 6.8 | 6955.6 |

Table 8: **(Latency profiling on a Nano 8G.)** TPOT and TTLT (256+256) are shown in *ms*. (OOM: out-of-memory)

| Method | W-bit | Prun. p% | Qwen-2.5-7B$^\diamond$ | | Mamba2-8B$^\ddagger$ | |
|---|---|---|---|---|---|---|
| | | | TPOT | TTLT | TPOT | TTLT |
| FP16 | 16 | 0% | OOM | OOM | OOM | OOM |
| TAO-HQQ | 4* | 0% | 129.8 | 38567.9 | - | - |
| **UniQL (Ours)** | 4 | 0% | 75.8 | 19447.0 | 81.6 | 21139.1 |
| | | 35% | 55.5 | 13792.0 | 58.2 | 14892.5 |

Table 9: **(The fused RoPE kernel.)** We profile TPOT on A6000 and report in *ms*.

| W-bit | Prun. p% | Fused RoPE | Llama-3.1 8B$^\diamond$ | Qwen-2.5 7B$^\diamond$ |
|---|---|---|---|---|
| 16 | 0% | - | 25.0 | 23.2 |
| | 25% | - | 20.2 | 18.7 |
| | | ✓ | 19.3 | 17.9 |
| 4 | 0% | - | 9.9 | 9.1 |
| | | ✓ | 9.0 | 8.3 |
| | 25% | - | 8.6 | 7.9 |
| | | ✓ | 7.7 | 7.1 |

Table 10: **(Accuracy by Components.)** We report the average accuracy for the models with different settings.

| W-bit | Prun. p% | +FT | +PTQ | +QSVD | Llama-3.1 8B$^\diamond$ | Qwen-2.5 7B$^\diamond$ |
|---|---|---|---|---|---|---|
| 16 | 0% | - | - | - | 74.0% | 72.4% |
| 16 | 25% | - | - | - | 67.0% | 62.1% |
| | | ✓ | - | - | 69.6% | 65.8% |
| 4 | 25% | - | ✓ | - | 55.2% | 56.1% |
| | | - | ✓ | ✓ | 65.0% | 60.7% |
| | | ✓ | ✓ | - | 60.2% | 61.0% |
| | | ✓ | ✓ | ✓ | 67.7% | 64.0% |

**Fused rotary positional embedding.** We compare latency with and without our fused RoPE. Since the positions are broken by our structured sorting, we have to collect the corresponding indices from the rotary positional embeddings, where we fuse in a kernel to minimize memory access. The fused RoPE kernel with index gathering yields a 10% latency reduction ($1.1\times$ speedup) for Llama-3.1-8B in 4-bit models at 0% and 25% compression, as depicted in Table 9.

**Masked LoRA fine-tuning.** We show that masked LoRA fine-tuning (FT) significantly benefits pruned models. As seen in Table 10, our method enhances accuracy by 2.6% (from 67.0% to 69.6%) and 3.7% (from 62.1% to 65.8%) for FP16 Llama-3.1-8B and Qwen-2.5-7B at 25% compression. For 4-bit models, it improves accuracy by 2.7% (from 65.0% to 67.7%) and 3.3% (from 60.7% to 64.0%) for Llama-3.1-8B and Qwen-2.5-7B at 25% compression.

**Quantization-aware decomposition.** We show the quantization-aware SVD decomposition (QSVD) is a key design to fill the performance gaps in Table 10. Low-bit quantization (*i.e.,* INT4) is sensitive to the numerical distribution in the quantization group. We decompose the weight matrix $\mathbf{W} = \mathbf{U\Sigma V}$ and combine the *long-tailed* eigenvalues $\mathbf{\Sigma}$ with $\mathbf{U}$, resulting in $\mathbf{W} = (\mathbf{U\Sigma})\mathbf{V}$, where a column of $\mathbf{U}$ is multiplied by the corresponding eigenvalue $\sigma_i$. Thus, $\sigma_i$ acts as the group's quantization scaling factor, as shown in Figure 4. We show this simple observation leads to significant performance gains 7.5% (from 60.2% to 67.7%) and 3% (from 61.0% to 64.0%) for 4-bit Llama-3.1-8B and Qwen-2.5-7B at the 25% pruning rate, respectively.

## 6 CONCLUSION

We present UniQL, a unified post-training compression framework that combines quantization and low-rank pruning to enable adaptive deployment of LLMs on edge. By supporting on-device configurable pruning and a one-shot cloud compression pipeline, UniQL addresses the key challenges posed by dynamic workloads. Through structured weight-sorting, quantization-aware decompositions, and fused rotary kernels, UniQL achieves substantial gains in memory and throughput across Transformers, SSMs, and hybrid models. Our results demonstrate that the compressed models can elastically adapt to runtime constraints.

## IMPACT STATEMENT

The UniQL framework has the potential to make large language models more accessible by enabling their deployment on edge devices with limited resources. This could broaden the scope of applications beyond high-end servers, potentially benefiting settings such as education, accessibility tools, or low-resource regions. At the same time, the increased availability of compact models raises concerns about potential misuse, including privacy risks and the generation of harmful or misleading content on widely distributed devices. Reducing the computational and memory footprint may also lessen the environmental costs of running large models, though the overall impact depends on the scale of adoption and usage patterns. We emphasize that UniQL itself does not mitigate societal risks associated with language model outputs, and responsible deployment practices remain necessary. By releasing our code and models, we aim to facilitate further research on efficient adaptation while encouraging the community to carefully consider both the benefits and risks of enabling lightweight edge deployment.

## ACKNOWLEDGMENTS

This work was supported in part by NSF CCF Grant No. 2107085, iMAGiNE - the Intelligent Machine Engineering Consortium at UT Austin, UT Cockrell School of Engineering Doctoral Fellowships, NSF CAREER Grant No. 2339084, Nvidia research gift, and Taiwan's NSTC Grant No. 111-2221-E-A49-148-MY3.

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

# A    DETAILED STRUCTURED SORTING ALGORITHMS

## A.1    MULTI-LAYER PERCEPTRON

Algorithm 1 outlines the structured sorting algorithm for an MLP. The MLP comprises up and down projections, with an optional gate projection. To obtain $\mathbf{S}_m$ for sorting weight matrices in an MLP layer, we gather the intermediate activation $\mathbf{X}_{\text{int}}$ from the calibration set and compute the channel correlation. We then average this correlation matrix over the set and compute the ridge leverage score. Using these scores, a sorting matrix $\mathbf{S}_m$ that sorts the output columns of $\mathbf{W}_u$ and $\mathbf{W}_g$ as $\mathbf{W}_u\mathbf{S}_m$ and $\mathbf{W}_g\mathbf{S}_m$. The rows of $\mathbf{W}_d$ are arranged as $\mathbf{S}_m^\top\mathbf{W}_d$ accordingly. We set ridge lambda $\lambda = 1$ in our experiments.

---

**Algorithm 1** Structured sorting for MLP.

---

1: **Input:** Up projection $\mathbf{W}_u \in \mathbb{R}^{D_{\text{h}} \times D_{\text{int}}}$, gate projection $\mathbf{W}_g \in \mathbb{R}^{D_{\text{h}} \times D_{\text{int}}}$, down matrix $\mathbf{W}_D \in \mathbb{R}^{D_{\text{int}} \times D_{\text{h}}}$, and hidden states $\mathbf{X}_{\text{h}}^i \in \mathbb{R}^{T \times D_{\text{h}}}$ from $N$ calibration samples $i = 1, ..., N$, and ridge intensity $\lambda$.
2: $\mathbf{X}_{\text{int}}^i = \sigma\left(\mathbf{X}_{\text{h}}^i\mathbf{W}_g\right) \odot \mathbf{X}_{\text{h}}^i\mathbf{W}_u$ , $i = 1, ..., N$
3: $\mathcal{C} = \frac{1}{N}\sum_{i=1}^{N}\mathbf{X}_{\text{int}}^{i\top}\mathbf{X}_{\text{int}}^i$, $\mathcal{C} \in \mathbb{R}^{D_{\text{int}} \times D_{\text{int}}}$   ▷ Average the correlation matrix over the samples
4: $s \leftarrow \text{diag}\left(\mathcal{C}(\mathcal{C} + \lambda I)^{-1}\right)$, $s \in \mathbb{R}^{D_{\text{int}}}$   ▷ Compute ridge leverage scores
5: $\mathbf{S}_m \leftarrow \mathbf{I}_{D_{\text{int}}}[:, \text{argsort}(s)]$, $\mathbf{S}_m \in \mathbb{R}^{D_{\text{int}} \times D_{\text{int}}}$   ▷ Get the sorting matrix based on the vector $s$
6: ▷ **Pseudo-inverse-free (Ours)**
7: **return** $(\mathbf{W}_u, \mathbf{W}_g, \mathbf{W}_d) \leftarrow \left(\mathbf{W}_u\mathbf{S}_m, \mathbf{W}_g\mathbf{S}_m, \mathbf{S}_m^\top\mathbf{W}_d\right)$ ▷ Output the structured sorted weights

---

## A.2    MULTI-HEAD SELF-ATTENTION: QUERY-KEY

Algorithm 2 describes the structured sorting for the query-key for MHSA. We obtain the activations for the $\mathbf{W}_q^j$ and $\mathbf{W}_k^j$, and compute the channel correlation $\mathcal{C}_q^j$ and $\mathcal{C}_k^j$. The sorting scores $s$ are calculated as $s = \|\mathcal{C}_q^{j\,1/2}\| \odot \|\mathcal{C}_k^{j\,1/2}\|$, and averaged over the calibration samples. To support the fused RoPE, we split the dimension of $s$ by half such that $[s_1, s_2] = s$, and apply sorting to $s_1 + s_2$. As such, we construct the final sorting matrix $\mathbf{S}_{qk}$ that sorts the output columns as $\mathbf{W}_q^j\mathbf{S}_{qk}^j$ and $\mathbf{W}_k^j\mathbf{S}_{qk}^j$.

---

**Algorithm 2** Structured sorting key-query for MHSA with $H$ heads.

---

1: **Input:** MHSA query matrices $\mathbf{W}_q \in \mathbb{R}^{D_{\text{h}} \times (H \times D_{\text{hd}})}$, key matrices $\mathbf{W}_k \in \mathbb{R}^{D_{\text{h}} \times (H \times D_{\text{hd}})}$, hidden states $\mathbf{X}_{\text{h}}^i \in \mathbb{R}^{T \times D_{\text{h}}}$ from $N$ calibration samples $i = 1, ..., N$, and the function of rotary positional embedding $\rho(\cdot)$.
  ▷ Apply sorting to each head independently
2: **for** $j = 1, \ldots, H$ **do**
3:    $\mathcal{C}_q^j = \frac{1}{N}\sum_{i=1}^{N}\rho(\mathbf{X}_{\text{h}}^i\mathbf{W}_q^j)^\top\rho(\mathbf{X}_{\text{h}}^i\mathbf{W}_q^j)$ , $\mathcal{C}_q^j \in \mathbb{R}^{D_{\text{hd}} \times D_{\text{hd}}}$   ▷ Query correlations
4:    $\mathcal{C}_k^j = \frac{1}{N}\sum_{i=1}^{N}\rho(\mathbf{X}_{\text{h}}^i\mathbf{W}_k^j)^\top\rho(\mathbf{X}_{\text{h}}^i\mathbf{W}_k^j)$ , $\mathcal{C}_k^j \in \mathbb{R}^{D_{\text{hd}} \times D_{\text{hd}}}$   ▷ Key correlations
5:    $s \leftarrow \|\mathcal{C}_q^{j\,1/2}\| \odot \|\mathcal{C}_k^{j\,1/2}\|$ , $s \in \mathbb{R}^{D_{\text{hd}}}$   ▷ Calculate the norm score
6:    ▷ **Symmetric sorting for fused RoPE kernel (Ours)**
7:    $[s_1, s_2] \leftarrow s$, $\{s_1, s_2\} \in \mathbb{R}^{D_{\text{hd}}/2}$   ▷ Split the norm score vector by half
8:    ▷ **Get the symmetric sorted indices**
9:    $\text{idx}_{\text{sym}} \leftarrow [\text{argsort}(s_1 + s_2), D_{\text{dh}}/2 + \text{argsort}(s_1 + s_2)]$, $\text{idx}_{\text{sym}} \in \mathbb{R}^{D_{\text{hd}}}$
10:   $\mathbf{S}_{qk} \leftarrow \mathbf{I}_{D_{\text{dh}}}[:, \text{idx}_{\text{sym}}]$, $\mathbf{S}_{qk} \in \mathbb{R}^{D_{\text{dh}} \times D_{\text{dh}}}$   ▷ Get the sorting matrix based on the vector $s$
11:   $(\mathbf{W}_q^j, \mathbf{W}_k^j) \leftarrow (\mathbf{W}_q^j\mathbf{S}_{qk}^j, \mathbf{W}_k^j\mathbf{S}_{qk}^j)$
12: **end for**
13: **return** $(\mathbf{W}_q, \mathbf{W}_k) \leftarrow \left([\mathbf{W}_q^1, \ldots, \mathbf{W}_q^H], [\mathbf{W}_k^1, \ldots, \mathbf{W}_k^H]\right)$   ▷ Concatenate the sorted heads

---

A.3 MULTI-HEAD SELF-ATTENTION: VALUE-OUTPUT

Algorithm 3 presents the quantization-aware SVD decomposition for $\mathbf{W}_v^i$ and $\mathbf{W}_o^i$ in MHSA. With the input correlation matrix $\mathcal{C}$, we perform joint decomposition by two consecutive SVD operations: $\mathcal{C}^{\frac{1}{2}}\mathbf{W}_v^i\mathbf{W}_o^i = \mathbf{U}_v\mathbf{U}\mathbf{\Sigma}\mathbf{V}^{\top}$. The SVD decomposition ranks the eigenvectors by eigenvalues. We decompose the weight matrix $\mathbf{W} = \mathbf{U}\mathbf{\Sigma}\mathbf{V}$ in a quantization-friendly fashion by merging the *long-tailed* eigenvalues $\mathbf{\Sigma}$ with $\mathbf{U}$, such that $\mathbf{W} = (\mathbf{U}\mathbf{\Sigma})V$.

---

**Algorithm 3** Structured sorting value-output for MHSA with $H$ heads.

---

1: **Input:** MHSA value matrices $\mathbf{W}_v \in \mathbb{R}^{D_{\mathrm{h}} \times (H \times D_{\mathrm{hd}})}$, output matrices $\mathbf{W}_o \in \mathbb{R}^{(H \times D_{\mathrm{hd}}) \times D_{\mathrm{h}}}$, hidden states $\mathbf{X}_{\mathrm{h}}^i \in \mathbb{R}^{T \times D_{\mathrm{h}}}$ from $N$ calibration samples $i = 1, ..., N$.
2: $\mathcal{C} = \mathbf{X}_{\mathrm{h}}^{i\top}\mathbf{X}_{\mathrm{h}}^i, \quad \mathcal{C} \in \mathbb{R}^{D_{\mathrm{hd}} \times D_{\mathrm{hd}}}$
    ▷ Apply sorting to each head independently
3: **for** $j = 1, \ldots, H$ **do**
4:     $(\mathbf{U}_v, \mathbf{\Sigma}_v, \mathbf{V}_v^{\top}) \leftarrow \mathrm{SVD}(\mathcal{C}^{1/2}\mathbf{W}_v^j)$
5:     $(\mathbf{U}, \mathbf{\Sigma}, \mathbf{V}^{\top}) \leftarrow \mathrm{SVD}(\mathbf{\Sigma}_v\mathbf{V}_v^{\top}\mathbf{W}_o^j)$
6:     $(\mathbf{W}_v^j, \mathbf{W}_o^j) \leftarrow (\mathcal{C}^{-1/2}\mathbf{U}_v\mathbf{U}\mathbf{\Sigma}, \mathbf{V}^{\top})$        ▷ **Quantization-aware SVD (Ours)**
7: **end for**
8: **return** $(\mathbf{W}_v, \mathbf{W}_o) \leftarrow ([\mathbf{W}_v^1, \ldots, \mathbf{W}_V^H], [\mathbf{W}_o^1, \ldots, \mathbf{W}_o^H])$        ▷ Concatenate the sorted heads

---

A.4 GROUP-QUERY ATTENTION: QUERY-KEY

Algorithm 4 describes the structured query-key sorting for GQA, featuring $H_s$ self-attention heads and $H_{kv}$ key-value heads, where $H_s > H_{kv}$. Firstly, we determine activations for $\mathbf{W}_k^j$ and then compute the channel correlation $\mathcal{C}_k^j$ for a key-value head $j$. The correlation $\mathcal{C}_k^j$ is shared among a group of self-attention heads. We then compute the channel correlation $\mathcal{C}_q^j$ for the group of self-attention heads, and sum the norm scores of the group. The sorting matrix $\mathbf{S}_{qk}$ is obtained similarly to MHSA to enable RoPE. In GQA, $\mathbf{S}_{qk}$ sorts the output columns of self-attention heads as $[\mathbf{W}_q^1\mathbf{S}_{qk}^j, \ldots, \mathbf{W}_q^{H_s/H_{kv}}\mathbf{S}_{qk}^j]$, and the key-value head as $\mathbf{W}_k^j\mathbf{S}_{qk}^j$.

---

**Algorithm 4** Structured sorting key-query for GQA with $H_s$ heads and $H_{kv}$ key-value heads.

---

1: **Input:** MHSA query matrices $\mathbf{W}_q \in \mathbb{R}^{D_{\mathrm{h}} \times (H_s \times D_{\mathrm{hd}})}$, key matrices $\mathbf{W}_k \in \mathbb{R}^{D_{\mathrm{h}} \times (H_{kv} \times D_{\mathrm{hd}})}$, hidden states $\mathbf{X}_{\mathrm{h}}^i \in \mathbb{R}^{T \times D_{\mathrm{h}}}$ from $N$ calibration samples $i = 1, ..., N$, and the function of rotary positional embedding $\rho(\cdot)$.
2: **for** $j = 1, \ldots, H_{kv}$ **do**
3:     $\mathcal{C}_k^j = \frac{1}{N}\sum_{i=1}^N \rho(\mathbf{X}_{\mathrm{h}}^i\mathbf{W}_k^j)^{\top}\rho(\mathbf{X}_{\mathrm{h}}^i\mathbf{W}_k^j), \; \mathcal{C}_k^j \in \mathbb{R}^{D_{\mathrm{hd}}}$        ▷ Key correlations
4:     $s \leftarrow [0, \ldots, 0], \; s \in \mathbb{R}^{D_{\mathrm{hd}}}$        ▷ Initialize $s$ with zeros
5:     **for** $\kappa = 1, \ldots, \frac{H_s}{H_{kv}}$ **do**
6:         $\mathcal{C}_q^{\kappa} = \frac{1}{N}\sum_{i=1}^N \rho(\mathbf{X}_{\mathrm{h}}^i\mathbf{W}_q^{\kappa})^{\top}\rho(\mathbf{X}_{\mathrm{h}}^i\mathbf{W}_q^{\kappa}), \; \mathcal{C}_q^{\kappa} \in \mathbb{R}^{D_{\mathrm{hd}}}$        ▷ Group query correlations
7:         $s = s + \|\mathcal{C}_q^{\kappa\,1/2}\| \odot \|\mathcal{C}_k^{j\,1/2}\|,$        ▷ Calculate the norm score
8:     **end for**
9:     $[s_1, s_2] \leftarrow s, \; \{s_1, s_2\} \in \mathbb{R}^{D_{\mathrm{hd}}/2}$        ▷ Split the norm score vector by half
10:    ▷ **Get the symmetric sorted indices**
11:    $\mathrm{idx}_{\mathrm{sym}} \leftarrow [\mathrm{argsort}(s_1 + s_2), D_{\mathrm{dh}}/2 + \mathrm{argsort}(s_1 + s_2)], \; \mathrm{idx}_{\mathrm{sym}} \in \mathbb{R}^{D_{\mathrm{hd}}}$
12:    $\mathbf{S}_{qk}^j \leftarrow \mathbf{I}_{D_{\mathrm{dh}}}[:, \mathrm{idx}_{\mathrm{sym}}], \; \mathbf{S}_{qk}^j \in \mathbb{R}^{D_{\mathrm{dh}} \times D_{\mathrm{dh}}}$        ▷ get the sorting matrix based on the vector $s$
13:    $([\mathbf{W}_q^1, \ldots, \mathbf{W}_q^{H_s/H_{kv}}], \mathbf{W}_k^j) \leftarrow ([\mathbf{W}_q^1\mathbf{S}_{qk}^j, \ldots, \mathbf{W}_q^{H_s/H_{kv}}\mathbf{S}_{qk}^j], \mathbf{W}_k^j\mathbf{S}_{qk}^j)$
14: **end for**
15: **return** $(\mathbf{W}_q, \mathbf{W}_k) \leftarrow ([\mathbf{W}_q^1, \ldots, \mathbf{W}_q^{H_s}], [\mathbf{W}_k^1, \ldots, \mathbf{W}_k^{H_{kv}}])$        ▷ Concatenate the sorted heads

---

## A.5 GROUP-QUERY ATTENTION: VALUE-OUTPUT

Algorithm 5 outlines structured value-output sorting for GQA. Since GQA has $H_s$ self-attention heads and $H_{kv}$ key-value heads, where $H_s > H_{kv}$, a single SVD decomposition is performed using the input correlation matrix $\mathcal{C}\mathbf{W}_v^j = \mathbf{U}_v\mathbf{\Sigma}_v\mathbf{V}^\top{}_v$. We also incorporate quantization-aware SVD for $\mathbf{W}_v^i$ by integrating $\mathbf{\Sigma}_v$ with $\mathbf{U}_v$ for $\mathbf{W}_v^j$. The $\mathbf{V}_v^\top$ is shared across attention heads such that $\mathbf{V}_v^\top\mathbf{W}_o^\kappa$ for $\kappa \in [1, \ldots, H_{kv}]$.

---

**Algorithm 5** Structured sorting value-output for GQA with $H_s$ heads and $H_{kv}$ key-value heads.

---

1: **Input:** MHSA value matrices $\mathbf{W}_v \in \mathbb{R}^{D_h \times (H_s \times D_{hd})}$, output matrices $\mathbf{W}_o \in \mathbb{R}^{(H_{kv} \times D_{hd}) \times D_h}$, hidden states $\mathbf{X}_h^i \in \mathbb{R}^{T \times D_h}$ from $N$ calibration samples $i = 1, ..., N$.
2: $\mathcal{C} = \mathbf{X}_h^{i\top}\mathbf{X}_h^i$,  $\mathcal{C} \in \mathbb{R}^{D_{hd} \times D_{hd}}$
3: **for** $j = 1, \ldots, H_{kv}$ **do**
4:     $(\mathbf{U}_v, \mathbf{\Sigma}_v, \mathbf{V}_v^\top) \leftarrow \text{SVD}(\mathcal{C}\mathbf{W}_v^j)$
5:     $\mathbf{W}_v^j \leftarrow \mathcal{C}^{-1}\mathbf{U}_v\mathbf{\Sigma}_v$                    ▷ **Quantization-aware SVD**
6:     **for** $\kappa = 1, \ldots, \frac{H_s}{H_{kv}}$ **do**
7:         $\mathbf{W}_o^\kappa \leftarrow \mathbf{V}_v^\top\mathbf{W}_o^\kappa$
8:     **end for**
9: **end for**
10: **return** $(\mathbf{W}_v, \mathbf{W}_o) \leftarrow \left([\mathbf{W}_v^1, \ldots, \mathbf{W}_V^{H_{kv}}], [\mathbf{W}_o^1, \ldots, \mathbf{W}_o^{H_s}]\right)$  ▷ Concatenate the sorted heads

---

## A.6 MAMBA: B-C STATE MATRICES

We show the algorithm that sorts the weights $\{\mathbf{W}_B^g, \mathbf{W}_C^g\}$ at Algorithm 6. For the SSM group $g$ with $H_m^g = \frac{H_m}{G_s}$ SSM heads in the group, we obtain the activations $\mathbf{B}^g$ and $\mathbf{C}^g$. $\mathbf{B}^g$ is discretized using the input-dependent variable $\Delta^g$ by a broadcasted outer product $(\Delta\mathbf{B})^g = \Delta^g \otimes \mathbf{B}^g$. As a result, we compute the channel correlation $\Delta\mathcal{C}_B^g$ and $\mathcal{C}_C^g$. The sorting scores $s \in \mathbb{R}^{D_s}$ are calculated as $s = \sum_{\tau=0}^{H_m^g}\|\tau\mathcal{C}_B^{g\,1/2}\| \odot \|\mathcal{C}_C^{g\,1/2}\|$, and averaged over the calibration samples. We construct the sorting matrix $\mathbf{S}_{BC}$ with $s$ that sorts the output columns of $\mathbf{W}_B^g$ and $\mathbf{W}_C^g$ as $\mathbf{W}_B^g\mathbf{S}_{BC}$ and $\mathbf{W}_C^g\mathbf{S}_{BC}$.

---

**Algorithm 6** Structured sorting B-C for Mamba with $H_m$ heads and $G_s$ SSM groups.

---

1: **Input:** B matrix $\mathbf{W}_B \in \mathbb{R}^{D_h \times (G_s \times D_s)}$ and C matrix $\mathbf{W}_C \in \mathbb{R}^{D_h \times (G_s \times D_s)}$, hidden states $\mathbf{X}_h^i \in \mathbb{R}^{T \times D_h}$ and input-dependent step size $\Delta^i \in \mathbb{R}^{T \times (H_m/G_s)}$ from $N$ calibration samples $i = 1, ..., N$, and the function of 1D causal convolution $\phi(\cdot)$.
2: **for** $j = 1, \ldots, G_s$ **do**
3:     $\mathbf{B}^{i,g} = \phi\left(\mathbf{X}_h^i\mathbf{W}_B^g\right)$, $\mathbf{B}^{i,g} \in \mathbb{R}^{T \times D_s}$
4:     $\mathbf{C}^{i,g} = \phi\left(\mathbf{X}_h^i\mathbf{W}_C^g\right)$, $\mathbf{C}^{i,g} \in \mathbb{R}^{T \times D_s}$
5:     $(\Delta\mathbf{B})^{i,g} = \Delta^{i,g} \otimes \mathbf{B}^{i,g}$, $(\Delta\mathbf{B})^{i,g} \in \mathbb{R}^{H_m^g \times T \times D_s}$     ▷ Broadcasted outer product
6:     ▷ Average over the calibration samples
7:     $\Delta\mathcal{C}_B^g = \frac{1}{N}\sum_{i=1}^N (\Delta\mathbf{B})^{i,g\top}(\Delta\mathbf{B})^{i,g}$, $\Delta\mathcal{C}_B^g \in \mathbb{R}^{(H_m/G_s) \times D_s \times D_s}$
8:     $\mathcal{C}_C^g = \frac{1}{N}\sum_{i=1}^N \mathbf{C}^{i,g\top}\mathbf{C}^{i,g}$, $\mathcal{C}_C^g \in \mathbb{R}^{D_s \times D_s}$
9:     ▷ Compute group correlations
10:     $s \leftarrow [0, \ldots, 0]$, $s \in \mathbb{R}^{D_{hs}}$                    ▷ Initialize $s$ with zeros
11:     **for** $k = 1, \ldots, \frac{H_m}{G_s}$ **do**
12:         $\mathcal{C}_B^k = (\Delta\mathcal{C}_B^g)^{k\top}(\Delta\mathcal{C}_B^g)^k$
13:         $s = s + \|\mathcal{C}_B^{k\,1/2}\| \odot \|\mathcal{C}_C^{j\,1/2}\|$,                ▷ Calculate the norm score
14:     **end for**
15:     $\mathbf{S}_{BC}^j \leftarrow \mathbf{I}_{D_s}[:, \text{argsort}(s)]$, $\mathbf{S}_{BC}^j \in \mathbb{R}^{D_s \times D_s}$  ▷ get the sorting matrix based on the vector $s$
16:     $(\mathbf{W}_B^j, \mathbf{W}_C^j) \leftarrow (\mathbf{W}_B^j\mathbf{S}_{BC}^j, \mathbf{W}_C^j\mathbf{S}_{BC}^j)$
17: **end for**
18: **return** $(\mathbf{W}_B, \mathbf{W}_C) \leftarrow \left([\mathbf{W}_B^1, \ldots, \mathbf{W}_B^{G_s}], [\mathbf{W}_C^1, \ldots, \mathbf{W}_C^{G_s}]\right)$ ▷ Concatenate the sorted states

---

## A.7 MAMBA: Z-X AND OUTPUT MATRICES

We describe the structured sorting for weights $\{\mathbf{W}_z^i, \mathbf{W}_x^i, \mathbf{W}_o^i\}$ in Algorithm 7. We collect the correlations $\mathcal{C}_{\mathcal{H}}^g$ from the SSM states $\mathcal{H}^i$ and average over the calibration samples. The ridge leverage score $\mathrm{diag}\left(\mathcal{C}_{\mathcal{H}}(\mathcal{C}_{\mathcal{H}} + \lambda I)^{-1}\right)$ is computed as the MLP layer. We create a column sorting matrix $\mathbf{S}_s \in \mathbb{R}^{D_{\mathrm{hd}} \times D_{\mathrm{hd}}}$ using ridge leverage scores, then arrange the output columns via $\mathbf{W}_z^i \mathbf{S}_s$ and $\mathbf{W}_x^i \mathbf{S}_s$. The input rows for $\mathbf{W}_o^i$ are sorted accordingly by $\mathbf{S}_s^\top \mathbf{W}_o^i$. We set ridge lambda $\lambda = 1$ in our experiments.

---

**Algorithm 7** Structured sorting z-x-o for Mamba with $H_m$ heads and $G_s$ SSM groups.

---

1: **Input:** x projection $\mathbf{W}_x \in \mathbb{R}^{D_{\mathrm{h}} \times (H_m \times D_{\mathrm{hd}})}$, z projection $\mathbf{W}_z \in \mathbb{R}^{D_{\mathrm{h}} \times (H_m \times D_{\mathrm{hd}})}$, out matrix $\mathbf{W}_o \in \mathbb{R}^{(H_m \times D_{\mathrm{hd}}) \times D_{\mathrm{h}}}$, and $\mathcal{H}^i \in \mathbb{R}^{H_m \times (T \times D_{\mathrm{s}}) \times D_{\mathrm{hd}}}$ from $N$ calibration samples $i = 1, ..., N$, and ridge intensity $\lambda$.
2: **for** $j = 1, \ldots, H_m$ **do**
3:     ▷ **State-aware (Ours)**
4:     $\mathcal{C} = \frac{1}{N} \sum_{i=1}^{N} \mathcal{H}^{i,j\top} \mathcal{H}^{i,j}$, $\mathcal{C} \in \mathbb{R}^{D_{\mathrm{hd}} \times D_{\mathrm{hd}}}$, $i = 1, ..., N$     ▷ Average over the samples
5:     $s \leftarrow \mathrm{diag}\left(\mathcal{C}(\mathcal{C} + \lambda I)^{-1}\right)$, $s \in \mathbb{R}^{D_{\mathrm{hd}}}$     ▷ compute ridge leverage scores
6:     $\mathbf{S}_s^j \leftarrow \mathbf{I}_{D_{\mathrm{hd}}}[:, \mathrm{argsort}(s)]$, $\mathbf{S}_s^j \in \mathbb{R}^{D_{\mathrm{hd}} \times D_{\mathrm{hd}}}$     ▷ get the sorting matrix based on the vector $s$
7:     $(\mathbf{W}_z^j, \mathbf{W}_x^j, \mathbf{W}_o^j) \leftarrow (\mathbf{W}_z^j \mathbf{S}_s^j, \mathbf{W}_x^j \mathbf{S}_s^j, \mathbf{S}_s^{j\top} \mathbf{W}_o^j)$
8: **end for**
9: **return** $(\mathbf{W}_z, \mathbf{W}_x, \mathbf{W}_o) \leftarrow \left([\mathbf{W}_z^1, \ldots, \mathbf{W}_z^{H_m}], [\mathbf{W}_x^1, \ldots, \mathbf{W}_x^{H_m}], [\mathbf{W}_o^1, \ldots, \mathbf{W}_o^{H_m}]\right)$     ▷ Concatenate the sorted heads

---

## B BROADER EVALUATION RESULTS

### B.1 COMPARISON WITH ADDITIONAL BASELINES

we include more common baselines in Table 11, with all methods evaluated in FP16 to ensure a fair and controlled setting. We follow the experimental setup used in MoDeGPT (Lin et al., 2025) and append our Llama-3.1-8B results at the 25% pruning rate to those reported in their work. The numbers, except for UniQL, are transcribed from MoDeGPT. The results show that UniQL consistently outperforms prior pruning-based methods at the 25% compression level, and UniQL-ft further boosts accuracy, achieving the strongest performance across all evaluated tasks.

Table 11: **(Comparison with Additional Baselines.)** All models are evaluated in FP16 for a fair and consistent comparison following the experimental setup used in MoDeGPT.

| Compress. % | Method | ARC-e | ARC-c | PIQA | WinoG. | HellaS. | Average |
|---|---|---|---|---|---|---|---|
| 0% | Dense | 77.69% | 53.58% | 80.63% | 72.69% | 79.16% | 72.75% |
| 25% | ShortGPT-Alpaca | 38.13% | 31.40% | 60.94% | 54.22% | 31.52% | 43.24% |
| | SliceGPT-Alpaca | 44.44% | 29.27% | 57.56% | 58.48% | 41.08% | 46.17% |
| | MoDeGPT-Alpaca | 67.05% | 41.13% | 75.52% | 69.61% | 66.49% | 63.96% |
| | UniQL | 70.37% | 46.33% | 74.16% | 71.82% | 70.12% | 66.56% |
| | UniQL-ft | 76.05% | 50.00% | 76.55% | 72.93% | 73.37% | 69.78% |

### B.2 EVALUATION ON THE MMLU DATASET

We test Llama-3.1-8B, Qwen-2.5-7B, and Nemotron-H-8B, and report five-shot accuracy on the MMLU dataset (Hendrycks et al., 2020) with a batch size of eight. The MMLU dataset is a large multitasking dataset, covering 57 subjects of varying difficulty. Our pruned models maintain competitive accuracy on the challenging dataset and outperform MoDeGPT (Lin et al., 2025) and SVD-LLM (Wang et al., 2025b). Nemotron-H, the Mamba-Transformer hybrid model, shows a large accuracy drop, but recovered with our low-cost masked fine-tuning. We compare our method with AWQ (Lin et al., 2024a) implemented in the TensorRT framework (NVIDIA, 2023; 2024) (TRT-AWQ) and HQQ (Badri & Shaji, 2023) in TorchAO (torchao, 2024) (TAO-HQQ). Our 4-bit models perform comparably to these state-of-the-art PTQ frameworks while offering broader model support.

Table 12: **(Five-shot MMLU.)** We compare UniQL against baselines under different settings. $*$ represent the FP16 embeddings and output layers as per the official implementation. $\perp$ denotes the GPTQ (Frantar et al., 2023) implemented on all models as an additional baseline.

| Method | +FT | +PTQ | W-bit | Prun. p% | Llama-3.1-8B$^\diamond$ | Qwen-2.5-7B$^\diamond$ | Nemotron-H-8B$^\dagger$ |
|---|---|---|---|---|---|---|---|
| FP16 | - | - | 16 | 0% | 65.6% | 74.2% | 67.6% |
| MoDeGPT | - | - | 16 | 15% | 59.5% | 23.1% | - |
| SVD-LLM | - | - | 16 | 15% | 28.4% | 49.9% | - |
| **UniQL (Ours)** | - | - | 16 | 15% | 60.2% | 55.9% | 37.5% |
| SVD-LLM | ✓ | - | 16 | 15% | 41.5% | 61.1% | - |
| **UniQL (Ours)** | ✓ | - | 16 | 15% | 59.2% | 59.9% | 56.1% |
| TRT-AWQ | - | ✓ | 4* | 0% | 63.0% | 72.5% | - |
| TAO-HQQ | - | ✓ | 4* | 0% | 62.9% | 72.5% | - |
| GPTQ$^\perp$ | - | ✓ | 4 | 0% | 61.5% | 70.5% | 64.0% |
| **UniQL (Ours)** | - | ✓ | 4 | 0% | 63.2% | 70.3% | 67.5% |
| SVD-LLM | ✓ | ✓ | 4* | 15% | 34.8% | 56.3% | - |
| **UniQL (Ours)** | ✓ | ✓ | 4 | 15% | 56.9% | 52.7% | 52.6% |

## B.3 EVALUATION ON CODING TASKS

We present the evaluation results on the MBPP+ (Austin et al., 2021; Gao et al., 2023) coding benchmark in Table 13. We apply UniQL to Llama-3.1-8B-Instruct and compare its performance against SVD-LLM (Wang et al., 2025b) and MoDeGPT (Lin et al., 2025) under various pruning ratios and bit-width configurations. The MBPP+ results obtained under batch size 1 and 0-shot settings following common practice. These results demonstrate UniQL's ability to maintain competitive performance compared to SVD-LLM and MoDeGPT while reducing model size.

Table 13: **(Evaluation results on the MBPP+ coding tasks.)** We compare UniQL with existing compression baselines under different pruning ratios and bit-width settings. Results are reported on the MBPP+ (instruct) benchmark using batch size 1 and 0-shot evaluation.

| Method | One-pass | +FT | W-bit | Prun. % | R.size ($\times$) | Llama-3.1-8B |
|---|---|---|---|---|---|---|
| FP16 | – | – | 16 | 0% | 0$\times$ | 75.4% |
| MoDeGPT | x | x | 16 | 15% | 0.15$\times$ | 42.3% |
| SVD-LLM | x | ✓ | 4 | 15% | 4.7$\times$ | 24.0% |
| UniQL (Ours) | ✓ | ✓ | 4 | 0% | 4$\times$ | 64.8% |
| | ✓ | ✓ | 4 | 15% | 4.7$\times$ | 54.2% |
| | ✓ | ✓ | 4 | 25% | 5.3$\times$ | 33.8% |

## C ADDITIONAL ABLATION STUDIES

### C.1 ABLATION STUDY ON CALIBRATION SETS

Table 14 presents the performance of Llama-3.1-8B under different combinations of calibration sets at a fixed 25% pruning rate. Following our setting, we report the average accuracy of five zero-shot downstream tasks. For each configuration, all hyperparameters and the number of calibration samples strictly follow the settings in Table 11 of our manuscript, ensuring a controlled and consistent comparison. Using the Alpaca dataset (Taori et al., 2023) for all stages results in the best average accuracy. We follow MoDeGPT and use WikiText2 as the calibration set for pruning-ratio allocation to ensure a fair comparison with prior work in all experiments.

Table 14: **(Ablation study on calibration sets.)** We report results for Llama-3.1-8B at a 25% pruning rate under different combinations of calibration sets used for weight-sorting, masked fine-tuning, and post-training quantization (PTQ).

| Prun. Ratio Alloc. | Weight-Sort. | Masked FT | PTQ | W-bit | Prun. Rate | Avg. Acc |
|---|---|---|---|---|---|---|
| – | – | – | – | 16 | 0% | 74.0% |
| wikitext2 | wikitext2 | wikitext2 | wikitext2 | 4 | 25% | 60.8% |
| wikitext2 | wikitext2 | alpaca | wikitext2 | 4 | 25% | 65.5% |
| wikitext2 | alpaca | alpaca | wikitext2 | 4 | 25% | 67.7% |
| alpaca | alpaca | alpaca | wikitext2 | 4 | 25% | 68.6% |

## C.2 ABLATION STUDY ON 3-BIT UNIQL

Our framework supports post-training quantization with various bit-widths, including 8, 6, 4, and even 3 bits. To support this claim, we include additional experiments exploring 3-bit UniQL in Table 15. The 3-bit precision is simulated by FP16 only for proof of concept purposes. These results demonstrate stable performance trends as the precision decreases, highlighting UniQL applicable to various bit-widths. Notably, even the 3-bit variant retains competitive accuracy across multiple models, underscoring the effectiveness of our weight-sorting and recovery fine-tuning procedure. Overall, these results highlight the flexibility of UniQL and confirm that it remains reliable even in resource-constrained, on-device deployment scenarios.

Table 15: **(Experimental results of 3-bit UniQL.)**

| Method | One pass | +FT | W bit | Prun. % | R.size (×) | Llama-2 7B | Llama-3.1 8B | Qwen-2.5 7B | Bamba-v2 9B | Nemotron-H 8B | Mamba2 8B |
|---|---|---|---|---|---|---|---|---|---|---|---|
| FP16 | – | – | 16 | 0% | 0× | 68.8% | 74.0% | 72.4% | 74.6% | 76.0% | 70.6% |
| MoDeGPT | x | x | 16 | 15% | 0.15× | 66.2% | 72.4% | 52.1% | – | – | – |
| SVD-LLM | x | ✓ | 4 | 15% | 4.7× | 63.2% | 60.6% | 66.8% | – | – | – |
| UniQL (Ours) | ✓ | ✓ | 4 | 0% | 4× | 67.6% | 73.6% | 72.4% | 75.1% | 73.3% | 69.3% |
| UniQL (Ours) | ✓ | ✓ | 4 | 15% | 4.7× | 65.6% | 71.4% | 68.1% | 70.3% | 70.5% | 65.8% |
| UniQL (Ours) | ✓ | ✓ | 4 | 25% | 5.3× | 63.5% | 67.7% | 64.0% | 67.4% | 64.7% | 61.8% |
| UniQL (Ours) | ✓ | ✓ | 3 | 0% | 5.3× | 62.8% | 64.5% | 67.4% | 67.8% | 71.3% | 67.8% |
| UniQL (Ours) | ✓ | ✓ | 3 | 15% | 6.2× | 60.2% | 63.5% | 63.0% | 64.4% | 67.8% | 64.0% |
| UniQL (Ours) | ✓ | ✓ | 3 | 25% | 7.1× | 58.9% | 59.4% | 58.7% | 61.6% | 62.1% | 60.3% |

## D PARETO-FRONT ANALYSIS

Figure 5 and 6 illustrate the Pareto-frontier trade-offs between accuracy and latency across a diverse set of Transformer, hybrid, and state-space models on A6000 and Nano 8G, respectively. Each sub-plot groups models by: (a) pure Transformers, (b) hybrid and SSM-based models, and (c) the union of both. We compare UniQL (W4A16, starred markers), GPTQ (Frantar et al., 2023) (W4A16, circular markers), and FP16 baselines (squares), with circle sizes indicating memory consumption on A6000. For the Nano 8G, we use HQQ (Badri & Shaji, 2023) that is supported in the TorchAO framework (torchao, 2024) (TAO-HQQ) as our baseline. Across all architectures, UniQL consistently yields better latency–accuracy trade-offs under the same memory budget, especially in the 2–4GB regime critical for edge deployment. For example, UniQL significantly improves latency for Qwen-2.5-7B and Mamba-2-8B while maintaining accuracy close to the FP16 baseline. Notably, UniQL achieves Pareto-dominant points for models like Llama-3.1-8B, outperforming GPTQ and TAO-HQQ in both latency and accuracy. Our analysis underscores UniQL's advantage for high-performance LLM inference under tight latency and memory constraints.

## E LAYER-WISE PRUNING RATES

We adopt the approach from (Lin et al., 2025; Men et al., 2024) to determine layer-wise pruning rates $r_l$ using Block Influence (BI) scores for specified global pruning rates. The BI score is given by

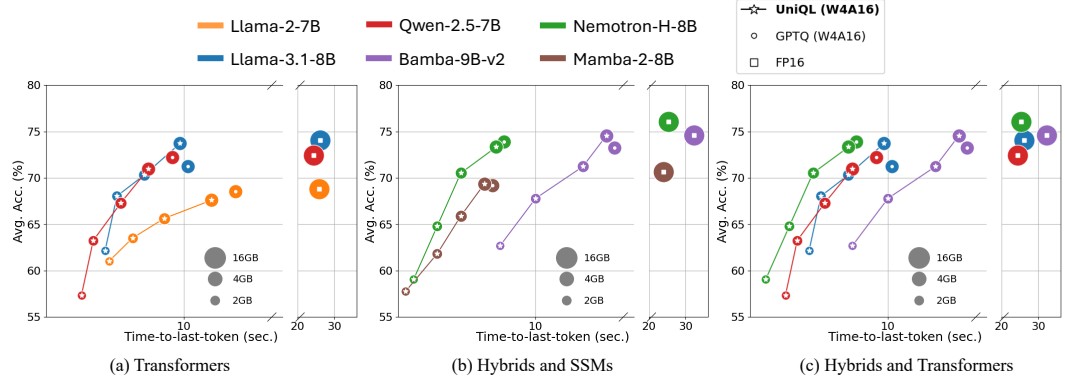

Figure 5: **(Pareto-front analysis on A6000.)** We evaluate the trade-off between average accuracy (%) and time-to-last-token (sec.) for various LLMs under different quantization and pruning configurations. Circle, square, and star markers denote GPTQ (W4A16), FP16, and our proposed UniQL (W4A16), respectively. Marker size indicates memory footprint.

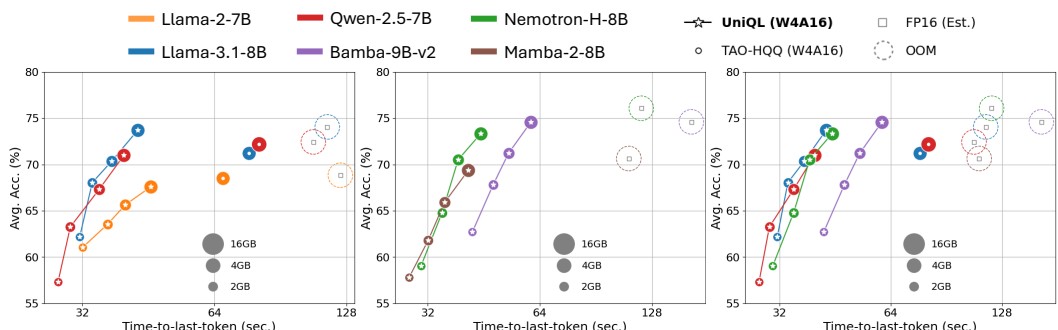

Figure 6: **(Pareto-front analysis on Nano 8G.)** We evaluate the trade-off between average accuracy (%) and time-to-last-token (sec.) for diverse LLMs with different quantization and pruning settings. Circle, square, and star markers represent TAO-HQQ (W4A16), FP16, and our UniQL (W4A16), respectively. Marker size reflects memory usage.

$s = 1 - \mathbb{E} \frac{\mathbf{x}_l^\top \mathbf{y}_l}{\|\mathbf{x}_l\|_2 \|\mathbf{y}_l\|_2}$, with $x_l$ and $y_l$ as the input and output of the block (*e.g.,* Transformer or Mamba block) at layer $l$. Using the closed-form solution from Lin et al. (2025), we smooth the layer-wise pruning rate allocations to obtain $P = [r_1^P, r_2^P, \ldots, r_L^P]$, such that $P = LP_{\text{avg}} \times \text{Softmax}(-\mathbf{s}/\varepsilon)$ where $s_i$ represents the importance score of layer $i$, and $P_{\text{avg}}$ denotes the target global sparsity. In our experiments, $\varepsilon$ is set to $0.1$, and we present the pruning rates for all models in Figure 7. Self-attention layers in hybrid models have low pruning rates, indicated by high BI scores in the Figure. In Bamba-9B-v2, layers 9, 18, and 27 are self-attention layers with lower pruning rates than nearby layers. Similarly, Nemotron-H-8B shows this pattern in layers 7, 18, 29, and 40. Pruning self-attention layers in hybrid models leads to significant accuracy drops.

# F IMPLEMENTATION DETAILS

## F.1 CALIBRATION SETS

Table 16 lists the calibration set, number of samples, and the sequence length we use in our experiments. We collect BI scores and assign layer-specific pruning rates using 128 samples with a sequence length of 2048 from wikitext-2 (Merity et al., 2017). Various global pruning rates, such as $P_{15}$, $P_{25}$, and $P_{35}$, are computed using the same setting. With 128 samples and a sequence length of 2048, we compute channel correlations from the Alpaca dataset (Taori et al., 2023), as detailed in (Lin et al., 2025). Our masked LoRA fine-tuning is conducted on the Alpaca dataset with a sequence

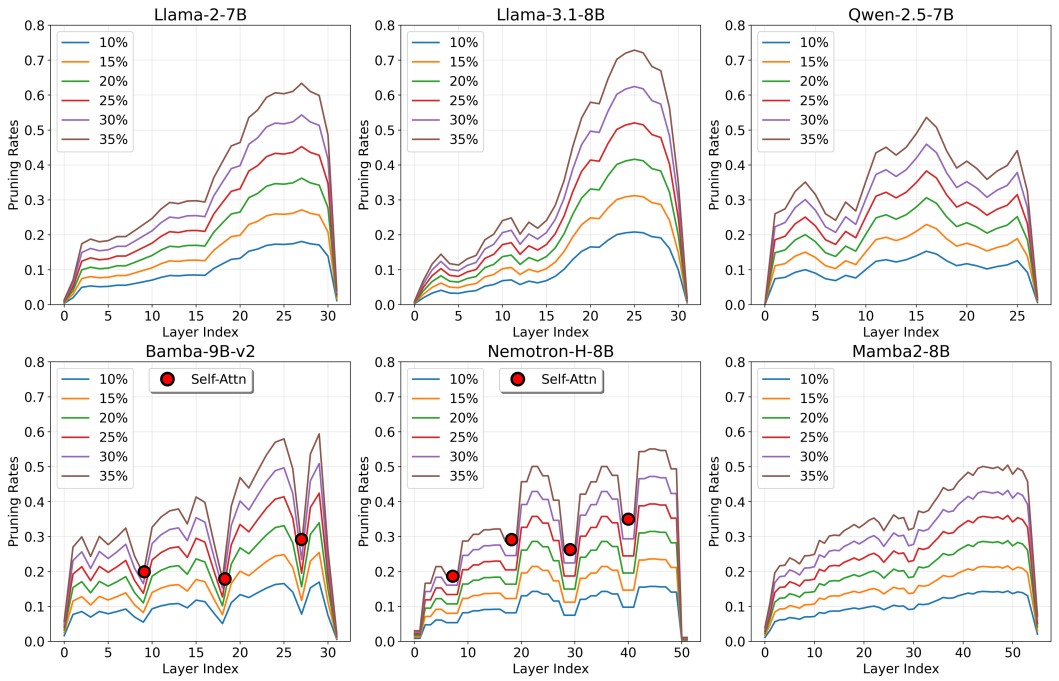

Figure 7: (**Layer-wise pruning rates.**)

length of 256 to reduce training memory usage. Lastly, we calibrate post-training quantization on the wikitext-2 dataset using 256 samples with a sequence length of 2048.

Table 16: (**Calibration sets.**)

|  | dataset | #data | seq. len. |
|---|---|---|---|
| Layer-wise prun. ratio alloc. | wikitext2 | 128 | 2048 |
| Structured weight-sorting | alpaca | 128 | 2048 |
| Masked LoRA fine-tuning | alpaca | 51800 | 256 |
| Post-training quantization | wikitext2 | 256 | 2048 |

## F.2    MASKED LORA FINE-TUNING

We list the hyper-parameters for our masked LoRA fine-tuning in Table 17. We follow the prior work to perform instruction tuning on the Alpaca dataset (Taori et al., 2023) for five epochs. Specifically, we adopt a relatively small sequence length of 256 to reduce training cost, and set the LoRA rank $r = 8$ with scaling factor $\alpha = 16$. A warmup of 100 steps is applied with the AdamW optimizer, and micro-batching is used to accommodate GPU memory limits. All models we experiment with are using the same hyperparameters, and we do not tune the parameters for the model and experiments.

Table 17: (**Hyperparameters for fine-tuning.**)

| Hyperparameter | Value |
|---|---|
| Learning rate | $1 \times 10^{-4}$ |
| Batch size | 32 |
| Micro batch size | 4 |
| Optimizer | AdamW |
| LoRA rank ($r$) | 8 |
| LoRA scaling ($\alpha$) | 16 |
| LoRA dropout | 0.05 |
| Warmup steps | 100 |
| Max sequence length | 256 |
| Training epochs | 5 |

## F.3    HADAMARD TRANSFORM FUSION

To enable the flexibility of the model size, our framework does not apply Hadamard rotations to the pruned channels. Importantly, the hidden dimension, *i.e.,*, the dimension propagated across layers,

remains unchanged. This design choice enables efficient on-device pruning for adaptive deployment, and avoid the mismatch shapes of the pre-fused Hadamard matrices after pruning the channels. We provide the detailed Hadamard fusion configurations for a Transformer block in Table 18 and a Mamba block in Table 19, where pruned channels are indicated with a "*". All other models follow the same Hadamard fusion pattern as these two examples. For Qwen-2.5-7B, we empirically find that applying Hadamard matrices degrades accuracy, so we remove all Hadamard matrices in our configuration.

Table 18: **(Hadamard matrix fusion for Transformer blocks.)**

| Operator | Input Had. | Output Had. |
|---|---|---|
| q_proj | ✓ Yes | ✗ No* |
| k_proj | ✓ Yes | ✗ No* |
| v_proj | ✓ Yes | ✗ No* |
| o_proj | ✗ No* | ✓ Yes |
| up_proj | ✓ Yes | ✗ No* |
| gate_proj | ✓ Yes | ✗ No* |
| down_proj | ✗ No* | ✓ Yes |

Table 19: **(Hadamard matrix fusion for Mamba blocks)**

| Operator | Input Had. | Output Had. |
|---|---|---|
| z_proj | ✓ Yes | ✗ No* |
| x_proj | ✓ Yes | ✗ No* |
| B_proj | ✓ Yes | ✗ No* |
| C_proj | ✓ Yes | ✗ No* |
| out_proj | ✗ No* | ✓ Yes |

## G    ENERGY PROFILING

To assess the practical efficiency of our quantization and pruning strategies, we conduct energy profiling on an A6000 GPU and an Orin Nano 8G, both of which are representative of cloud and edge platforms. On Orin Nano, each request is prefilled with 512 tokens and generates 512 new tokens, where we record the total energy consumption in Joules-per-request (J/req.). As shown in Table 20, full-precision (FP16) models exceed the device's 8 GB memory limit, resulting in out-of-memory (OOM) errors during inference. In contrast, quantized methods substantially reduce energy consumption while maintaining deployability. Without pruning, UniQL (W4A16) reduces the energy per request to 208.23 J and 224.56 J on Qwen-2.5-7B◇ and Mamba2-8B‡, respectively. When combined with structured pruning ($p=35\%$), the energy further decreases to 143.12 J and 153.64 J.

Table 20: **(Energy profiling on Nano.)** Joules-per-request (J/req.) is reported. Each request is prefilled with 512 tokens and 512 generated tokens. Lower is better ($\downarrow$). * represent the FP16 embeddings and output layers as per the official implementation.

| Method | W-bit | Prun. p% | Qwen-2.5-7B◇ J/req. $\downarrow$ | Mamba2-8B‡ J/req. $\downarrow$ |
|---|---|---|---|---|
| FP16 | 16 | 0% | OOM | OOM |
| TAO-HQQ | 4* | 0% | 381.13 | - |
| **UniQL (Ours)** | 4 | 0% | 208.23 | 224.56 |
|  |  | 35% | 143.12 | 153.64 |

On cloud GPUs, we evaluate energy efficiency in terms of *tokens-per-Gigawatt* to align with the industrial computing power metric. In Figure 8, we visualize the energy efficiency of a Transformer model (*i.e.,* Llama-3.1-8B) and a Hybrid model (*i.e.,* Nemotron-H-8B) on an A6000 GPU with 48GB memory. Each request is prefilled with 1024 tokens and generates 1024 new tokens. Under different batch sizes, we report the total number of tokens for a Gigawatt per second. Nemotron-H adopts SSM blocks to reduce the memory needs for KV cache. Also, UniQL consistently achieves higher throughput-per-energy across both Transformer-based and SSM-based architectures. This establishes UniQL as an effective deployment framework for both resource-constrained and energy-aware scenarios.

---

*The embedding and output layers use FP16 according to the official implementation.

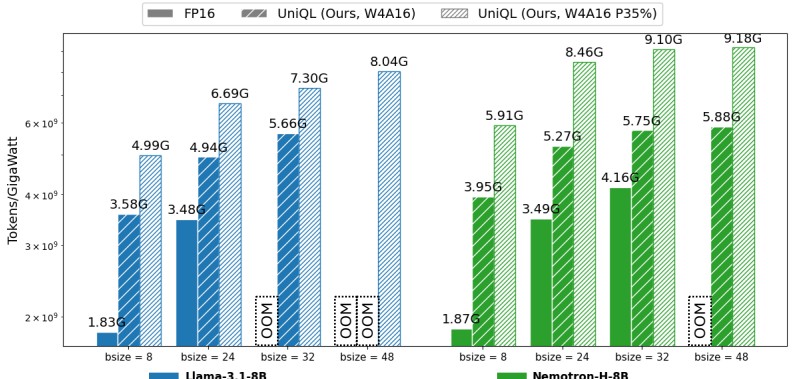

Figure 8: **(Energy efficiency analysis on A6000.)** Nemotron-H incorporates SSM blocks to decrease KV cache memory requirements. UniQL continually offers superior energy efficiency for both Transformer and SSM architectures.

