# OpenReview forum: "UniQL: Unified Quantization and Low-rank Compression for Adaptive Edge LLMs"
_ICLR.cc/2026/Conference — ICLR 2026 Poster_

### Official Review · Reviewer_ZDeG · 2025-10-23

**Soundness:** 2
**Presentation:** 3
**Contribution:** 2
**Rating:** 4
**Confidence:** 4

**Summary:**

The paper targets dynamic, resource-constrained edge deployment of LLMs. It proposes a one-shot, post-training pipeline that produces a single INT4 model which can be adaptively pruned on-device at different rates depending on runtime constraints. Overall, the paper is a well-executed systems integration that addresses a timely deployment problem and demonstrates tangible engineering wins. However, for ICLR, the lack of clear algorithmic novelty and the absence of a rigorous case for “why combine pruning with quantization instead of simply going lower-bit” (especially W3) are significant gaps. Clarifying the scope of Hadamard/rotation, aligning it with pruning, and adding stronger/fairer baselines and ablations would strengthen the submission.

**Strengths:**

1. Practicality and scope: A unified, one-shot post-training pipeline that supports Transformers, SSMs, and hybrids, and enables on-device adaptive pruning without retraining is very relevant for edge deployment.
2. Solid engineering: the fused RoPE kernel, quantization-aware SVD, and avoidance of pseudo-inverses show careful engineering; the export path quantizes even embeddings/LM head to 4-bit, reducing footprint beyond common W4A16 libraries.

**Weaknesses:**

1. Most core techniques (importance-driven structured pruning, Hadamard/rotation-based quantization smoothing, W4 PTQ) are known. The main novelty is system integration and some engineering refinements (QSVD, fused RoPE, SSM-aware sorting). This level of originality feels marginal for ICLR unless paired with stronger conceptual/theoretical advances or broader evidence.

2. No justification for “quantization + pruning” vs “lower-bit quantization alone”: The paper does not compare W4 + k% pruning to a lower-bit alternative (e.g., W3) under matched memory/latency. Prior results (e.g., QuaRot with 3-bit) suggest that W3 may match or exceed the accuracy of W4 + 25% pruning at similar or lower budgets. Without this key comparison, the necessity of combining pruning with quantization remains unproven.

3. Weak PTQ baselines : PTQ comparisons are largely against framework built-ins (TRT-AWQ, TAO-HQQ) and a basic GPTQ variant; missing stronger recent baselines such as AWQ/QServe, Quarot/SpinQuant, etc.

4. The scope of rotations is not clearly specified: If rotations are applied on input channel axes of layers whose channels will later be pruned (e.g., O_proj or MLP down-proj input channels), this mismatch can harm pruning efficacy. There is no ablation on restricting rotations to non-pruned axes or aligning pruning boundaries with quantization/rotation groups.

**Questions:**

1. Please provide matched-budget comparisons of W4 + {25}% pruning versus W3 (with Hadamard/rotation, e.g., QuaRot-style) on the same models, reporting accuracy–memory Pareto curves. Under what budgets does “W4 + pruning” strictly dominate “W3 alone”?

2. Which layers/axes are rotated (Q/K/V/O proj in attention; MLP up/gate/down; SSM B/C/Z/X/O)? Are rotations applied to channel dimensions that will be pruned on-device?

---

> ### Author Response · Authors · 2025-11-20
> **Response to Reviewer ZDeG (Part 1)**
>
> We thank the reviewer for their time and effort in evaluating our paper. Below, we highlight our main contributions and provide detailed responses to each question and comment. We look forward to a constructive discussion with the reviewer and to further clarifying and justifying the assigned ratings.
>
> > Most core techniques (importance-driven structured pruning, Hadamard/rotation-based quantization smoothing, W4 PTQ) are known. The main novelty is system integration and some engineering refinements (QSVD, fused RoPE, SSM-aware sorting)
>
> While we remain open to constructive discussion with the reviewer, we highlight our main contributions below which do not support the reviewer’s claim of lack of novelty.
>
> Our goal is to present a novel compression framework that enables flexible low-bitwidth models for edge devices with unified memory, applicable to Transformers, State Space Models (SSMs), and hybrid architectures. We introduce several new algorithmic components: a pseudo-inverse-free, quantization-aware SVD, state-aware weight sorting, and a fused rotary embedding kernel, all specifically designed for the proposed unified framework.
>
> To the best of our knowledge, our work is the first to: (1) address flexible model compression under a **novel unified-memory setting** for edge devices, (2) systematically combine quantization and structured pruning with **several new algorithms**, and (3) provide a unified framework applicable to a **wide spectrum of architectures**.
>
> In summary, from the problem formulation to the algorithmic and framework design, our work is **far from incremental** or **purely engineering-driven**. To further clarify our contributions, we present a comparison in Table R19 to help reviewers better understand our contributions below.
>
> We would appreciate it if the reviewer could point to concrete recent work that motivates their comment and engage in a constructive discussion to help us further improve our work.
>
> [Table R19: UniQL compared to prior quantization methods]
> | Method  | Quant. | Structured Prun. | Joint Design (Quant.+Prun.) | Single-pass | On-device Prun. | Supported Arch. |
> |-------------|----------|------------------------|--------------------------------|-------------|------------------|------------------|
> | **UniQL (Ours)** | ✔️ Yes | ✔️ Yes | ✔️ Yes | ✔️ Yes | ✔️ Yes | Transformers / SSMs / Hybrid |
> | Quarot                | ✔️ Yes  | ✘ No   | ✘ No    | ✘ No  | ✘ No   | Transformers only |
> | SpinQuant          | ✔️ Yes  | ✘ No   | ✘ No    | ✘ No  | ✘ No   | Transformers only |
> | QServe               | ✔️ Yes  | ✘ No   | ✘ No    | ✘ No  | ✘ No   | Transformers only |
>
>
> > Please provide matched-budget comparisons of W4 + {25}% pruning versus W3 (with Hadamard/rotation, e.g., QuaRot-style) on the same models, reporting accuracy–memory Pareto curves. Under what budgets does “W4 + pruning” strictly dominate “W3 alone”?
>
> The comparison between “quantization + pruning” and “lower-bit quantization alone” is **not the primary focus** of this work. Our goal is to present a novel compression framework that generates flexible n-bit models for edge devices with unified memory. In fact, our framework supports post-training quantization with any bit-width such as 8, 6, 4, 3. To support our claim, we provide 3-bit UniQL with adaptive pruning rate in Table R20. The 3-bit precision is simulated by FP16 only for proof of concept purposes.
>
> [Table R20: Experimental results of 3-bit UniQL]
> | Method | Sing-pass | +FT | W-bit | Prun. % | R.size (×) | Llama-2-7B | Llama-3.1-8B | Qwen-2.5-7B | Bamba-v2-9B | Nemotron-H-8B | Mamba2-8B |
> |:-------------------|:--:|:--:|:---:|:-----:|:------:|:--------:|:-------:|:------:|:-------------:|:-----------:|:--------:|
> | FP16              | - | -  | 16 | 0%  | 0×      | 68.8% | 74.0% | 72.4% | 74.6% | 76.0% | 70.6% |
> | MoDeGPT      |x | x |  16 | 15%| 0.15× |  66.2% | 72.4% | 52.1% | – | – | – |
> | SVD-LLM       |x | ✓ |  4 | 15% | 4.7×  | 63.2% | 60.6% | 66.8% | – | – | – |
> | UniQL (Ours) |✓ | ✓ |  4 | 0%   | 4×     | 67.6% | 73.6% | 72.4% | 75.1% | 73.3% | 69.3% |
> |                       |✓ | ✓ |  4 | 15% | 4.7× | 65.6% | 71.4% | 68.1% | 70.3% | 70.5% | 65.8% |
> |                       |✓ | ✓ |  4 | 25% | 5.3× | 63.5% | 67.7% | 64.0% | 67.4% | 64.7% | 61.8% |
> | UniQL (Ours) |✓ | ✓ |  3 | 0%   | 5.3× | 62.8% | 64.5% | 67.4% | 67.8% | 71.3% | 67.8% |
> |                       |✓ | ✓ |  3 | 15% |  6.2x | 60.2% | 63.5% | 63.0% | 64.4% |  67.8% | 64.0% |
> |                       |✓ | ✓ |  3 | 25% |  7.1x | 58.9% | 59.4% | 58.7% | 61.6% |  62.1% | 60.3% |

---

> > ### Author Response · Authors · 2025-11-20
> > **Response to Reviewer ZDeG (Part 2)**
> >
> > > missing stronger recent baselines such as AWQ/QServe, Quarot/SpinQuant, etc.
> >
> > We kindly remind the reviewers that our work focuses on **weight-only** (W4A16) post-training quantization (PTQ), and therefore weight-activation PTQ methods such as SpinQuant (W4A4) [12], Quarot (W4A4) [13], and QServe (W4A8) [15]  fall outside the scope of this work.
> >
> > To clarify the appropriate use cases for **weight-only** (W4A16) versus **weight-activation** quantization, we provide a comparison in Table R19 and R21 to help reviewers better understand the positioning and contributions of our method. For completeness, we summarize the most common configurations used in post-training quantization.
> >
> > [Table R21: Comparison between weight-only and weight-activation quantization]
> > | Quant. type        | W-bit     | A-bit     | Pros                                   | Cons                               | Application |
> > |--------------------|-----------|-----------|----------------------------------------|------------------------------------|-------------|
> > | Weight-only | 4 / 6 / 8 | 16        | Higher accuracy; reduces memory load   | No gain in compute-bound cases     | Edge or small-batch applications with high accuracy needs |
> > | Weight-activation | 4 / 6 / 8 | 4 / 6 / 8 | Reduces both memory and compute cost   | Usually loss more in accuracy  | Cloud serving with more accuracy tolerance |
> >
> >
> > In Qserve, the paper mentioned by the reviewer, the authors also mention that W4A4 methods [13, 14] introduce a notable accuracy degradation, while exhibiting 20-25% lower throughput than its W4A16 and W8A8 counterparts in TensorRT-LLM. The results can be seen in Figure 2(b), Table 4 and Figure 17 in the Qserve paper.
> >
> > To address reviewers' concern, we profile the generation throughput comparison under our edge setting (i.e., batch size 1) for Llama-2-7B and compare it with QServe on A100. As seen in Tables R22 and R23, UniQL provides superior throughput and accuracy.
> >
> > [Table R22: Generation throughput comparison for Llama-2-7B on A100]
> > | Method        | Precision | Throughput (tokens/sec) |
> > |---------------|:---------:|-------------------------:|
> > | TRT-LLM FP16  |   FP16    | 102.56                   |
> > | UniQL W4A16   |  W4A16    | 129.19                   |
> > | QServe (QoQ)  |  W4A8     | 111.41                   |
> >
> >
> > [Table R23: Accuracy comparison for Llama-2-7B across quantization methods]
> >
> > | Precision       | Method | Prun. %| PQ    | ARC-e | ARC-c | HS    | WG    | Avg.  |
> > |------------------|--------|-------|-------|-------|-------|-------|-------|-------|
> > | FP16                  | -      | 0 %| 79.05 | 74.58 | 46.25 | 76.05 | 68.98 | 68.98 |
> > | W4A4                 | Quarot | 0%  | 76.77 | 69.87 | 40.87 | 72.16 | 63.77 | 64.69 |
> > | W4A4 g128        | Atom    | 0%  | 75.14 | 52.99 | 38.40 | 69.37 | 62.75 | 59.73 |
> > | W4A8KV4          | QoQ     | 0%  | 78.07 | 73.11 | 45.05 | 74.12 | 67.48 | 67.57 |
> > | W4A8KV4 g128 | QoQ     | 0%  | 78.07 | 73.32 | 44.80 | 74.98 | 68.59 | 67.95 |
> > | W4A16               | UniQL  | 0%  | 78.67 | 72.69 | 42.75 |  75.07 | 68.82 | 67.60 |
> > |                            |             | 15% | 76.17 | 69.23   | 40.96 | 72.88  | 68.90  | 65.63  |
> > |                            |             | 25% | 74.32 | 66.12  | 39.59   | 70.30 |  67.32  | 63.53 |

---

> ### Author Response · Authors · 2025-11-20
> **Response to Reviewer ZDeG (Part 3)**
>
> > The scope of rotations is not clearly specified: If rotations are applied on input channel axes of layers whose channels will later be pruned (e.g., O_proj or MLP down-proj input channels), this mismatch can harm pruning efficacy. There is no ablation on restricting rotations to non-pruned axes or aligning pruning boundaries with quantization/rotation groups. Which layers/axes are rotated (Q/K/V/O proj in attention; MLP up/gate/down; SSM B/C/Z/X/O)? Are rotations applied to channel dimensions that will be pruned on-device?
>
> As the reviewer notes, our framework does not apply Hadamard rotations to the pruned channels. We note that the hidden dimension, i.e., the dimension propagated from layer to layer, remains unchanged. This design choice allows us to perform on-device pruning for adaptive deployment. To address the reviewer’s concern, we provide the detailed configuration of our Hadamard fusion for Transformer blocks in Table R24 and Mamba blocks in Table R25. The channels to be pruned are marked with a “*”. Other models follow the same pattern for Hadamard fusion with Llama-3.1-8B and Mamba2-8B. For Qwen-2.5-7B, we empirically find that applying Hadamard matrices degrades accuracy, so we remove all Hadamard matrices in our configuration.
>
> [Table R24: Hadamard matrix fusion for Transformer blocks]
> | Operator   | Input Had. | Output Had. |
> |------------|------------|-------------|
> | q_proj     |    ✔️ Yes        |      ✘ No *       |
> | k_proj     |     ✔️ Yes       |      ✘ No *       |
> | v_proj     |     ✔️ Yes       |      ✘ No *      |
> | o_proj     |       ✘ No *     |      ✔️ Yes       |
> | up_proj    |      ✔️ Yes      |      ✘ No *       |
> | gate_proj  |    ✔️ Yes        |      ✘ No *      |
> | down_proj |     ✘ No *      |        ✔️ Yes     |
>
> [Table R25: Hadamard matrix fusion for Mamba blocks]
> | Operator   | Input Had. | Output Had. |
> |------------|------------|-------------|
> | z_proj     |    ✔️ Yes        |      ✘ No *      |
> | x_proj     |     ✔️ Yes       |      ✘ No *      |
> | B_proj     |     ✔️ Yes       |      ✘ No *      |
> | C_proj     |       ✔️ Yes     |        ✘ No *     |
> | out_proj  |    ✘ No *       |      ✔️ Yes        |

---

> > ### Author Response · Authors · 2025-11-25
> > **Discussion period reminder**
> >
> > Dear Reviewer ZDeG,
> >
> > We hope our rebuttal has addressed your questions and concerns. We look forward to a constructive discussion and would appreciate any justification for your current rating before the discussion period closes.

---

### Official Review · Reviewer_8Vsv · 2025-10-28

**Soundness:** 3
**Presentation:** 3
**Contribution:** 3
**Rating:** 8
**Confidence:** 3

**Summary:**

This paper proposes UniQL, a unified post-training quantization and low-rank compression framework designed to support adaptive deployment of large language models on resource-constrained edge devices. UniQL achieves integrated compression for Transformer, SSM, and hybrid models through structured weight ranking, quantization-aware SVD decomposition, state-aware SSM compression, and the integration of RoPE kernels. The framework performs one-time compression in the cloud and supports dynamic on-device pruning (up to 35%) based on current load. Experiments demonstrate that UniQL achieves significant memory reduction and inference acceleration across multiple models and hardware platforms, with manageable accuracy loss.

**Strengths:**

Wide Model Architecture Coverage: This approach systematically supports post-training quantization and structured pruning for Transformer, SSM, and hybrid models for the first time.

Strong On-Device Adaptability: This approach dynamically adjusts model size based on memory and compute resources after deployment to adapt to the dynamic load of edge devices.

High Compression Efficiency: This approach significantly accelerates the compression process (up to 22x faster than MoDeGPT) by avoiding pseudo-inverses and optimizing SVD decomposition.

System-Level Optimization: This approach integrates RoPE kernels and quantization-aware decomposition to significantly improve inference speed and reduce quantization error.

Extensive Experimentation: This approach demonstrates its effectiveness and versatility across a variety of models (e.g., Llama, Qwen, Mamba) and hardware (e.g., A6000, Nano 8G).

**Weaknesses:**

Accuracy-compression tradeoff not yet optimal: At high pruning rates (e.g., 35%), the accuracy of some models (e.g., Mamba2) drops significantly (down to 57.7%).

Sensitive to calibration data: Structured ordering and quantization depend on the calibration set; their robustness to different data distributions is not analyzed.

Lack of comparison with unstructured methods: No comparison with popular unstructured pruning methods (e.g., SparseGPT) or hybrid sparse methods is provided.

Device-side pruning overhead not quantified: While device-side pruning is supported, its runtime overhead (e.g., memory rearrangement, index lookup) is not analyzed.

Limited interpretability: No visualization or interpretable analysis is provided for the "state-aware" or "quantization-aware" mechanisms.

**Questions:**

Calibration Data Sensitivity:
How sensitive is UniQL to the choice of calibration data? Have you tested its robustness across domains (e.g., code, math, dialogue) or with out-of-distribution samples?

Comparison with Unstructured Pruning:
How does UniQL compare with unstructured pruning methods like SparseGPT, especially in terms of accuracy-efficiency trade-offs and hardware friendliness?

---

> ### Author Response · Authors · 2025-11-20
> **Response to Reviewer 8Vsv (Part 1)**
>
> We thank the reviewer for their positive feedback. Detailed responses to reviewer’s questions and concerns are provided below. We look forward to constructive discussions to further improve our work and justify the rating.
>
> > Accuracy-compression tradeoff not yet optimal: At high pruning rates (e.g., 35%), the accuracy of some models (e.g., Mamba2) drops significantly (down to 57.7%).
>
> Given the shorter compression time of our algorithm, we note that our compressed models achieve higher accuracy than state-of-the-art methods such as SVD-LLM, MoDeGPT. In Table R11, we compare all state-of-the-art methods for Llama-2-7B at 35% pruning rate to support our claim. We report average accuracy of zero-shot downstream tasks in the Table. Moreover, our support for compressing both Mamba and hybrid models further distinguishes our work from prior structured pruning and quantization studies [4, 5, 6, 8, 9].
>
> [Table R11: Comparing accuracy at 35% pruning rate]
> | Method     | FT | W-bit | Accuracy (%) |
> |----------------|:--:|:-----:|--------------:|
> | FP16          | -  | 16    | 68.8%         |
> | MoDeGPT  | ✗  | 16     | 56.9%        |
> | SVD-LLM  | ✗  | 16     | 45.0%        |
> | UniQL  | ✗  | 16     | 60.0%        |
> | SVD-LLM   | ✓  | 4     | 56.1%        |
> | UniQL    | ✓ | 4     | 61.0%        |
>
> > Sensitive to calibration data: Structured ordering and quantization depend on the calibration set; their robustness to different data distributions is not analyzed. How sensitive is UniQL to the choice of calibration data?
>
> We follow MoDeGPT and use WikiText2 as the calibration set for pruning-ratio allocation to ensure a fair comparison. We provide the calibration set used in our experiments in the appendix for reproducibility. To address your concern, Table R15 reports results under different calibration set combinations for Llama-3.1-8B at 25% pruning rate.For each dataset change, all hyperparameters and the number of calibration samples follow the settings specified in Table 11 of our manuscript.
>
> [Table R15: Ablation study on calibration sets]
> | Pruning Ratio Alloc. | Weight-Sorting | Masked Fine-Tuning | PTQ | Bit-width | Prun. Rate      | Avg. Acc |
> |----------------------------|----------------|---------------------|-----------|----------|----------|----------|
> | -                               | -                 | -                       | -              | 16 | 0%   | 74.0%  |
> | wikitext2                  | wikitext2    | wikitext2          | wikitext2  | 4  | 25% | 60.8%  |
> | wikitext2                  | wikitext2    | alpaca             | wikitext2  | 4  | 25% |  65.5%    |
> | wikitext2                  | alpaca       | alpaca             | wikitext2   | 4 | 25% |  67.7%  |
> | alpaca                     | alpaca       | alpaca             | wikitext2      | 4  | 25% |  68.6%  |
>
>
>
> > Device-side pruning overhead not quantified: While device-side pruning is supported, its runtime overhead (e.g., memory rearrangement, index lookup) is not analyzed.
>
> To achieve the best inference latency, we capture a static CUDA graph; therefore, we prune the model size before deployment under the current framework.
>
> In practice, we load and prune the model on the CPU. Specifically, we utilize the available main memory and swap space to prune low bit-width models on-device. After pruning, the model is loaded into the remaining main memory shared between the CPU and GPU. Empirically, this entire process takes 2 minutes using the 6-core Arm Cortex-A78AE 64-bit CPU on Nano 8G. We note that several optimization techniques can be applied to this process, such as parallelizing computations across all layers [6] and concurrently pruning the model to a new size on the CPU while it is being deployed on the GPU.
>
> Our work focuses on algorithmic framework design and, to the best of our knowledge, is the first to address the unified memory challenge on modern edge devices. Since the dynamic memory management for CUDA graphs is not yet supported by major inference frameworks such as TensorRT-LLM [1] and SGLang [11], we leave further improvements to future work.

---

> ### Author Response · Authors · 2025-11-20
> **Response to Reviewer 8Vsv (Part 2)**
>
> > Limited interpretability: No visualization or interpretable analysis is provided for the "state-aware" or "quantization-aware" mechanisms.
>
> We thank reviewers for their suggestions, and have updated our manuscript and highlighted the corresponding lines in the updated version. We visualize the quantization-aware SVD  in Figure 4 (right), and Algorithm 3 and 5. The state-aware decomposition can be found at line 266 and in Algorithm 7.
>
> > Have you tested its robustness across domains (e.g., code, math, dialogue) or with out-of-distribution samples?
>
> Yes, we provide the MMLU results in Table 12 of the appendix. MMLU covers a diverse set of tasks spanning STEM, math, law, medicine, humanities, and other professional domains, enabling us to evaluate the model’s robustness across a wide range of out-of-distribution scenarios. We also note that UniQL is a one-pass framework, so we did not tune the calibration set or LoRA parameters for MMLU.
>
> Additionally, We present the evaluation results on coding (MBPP+) tasks [10] in Table R16, respectively. We apply UniQL to Llama-3.1-8B-Instruct, and compare their performance against SVD-LLM. To assess cross-domain generalization, we evaluate on both coding and math benchmarks. For coding, we report MBPP+ (instruct variant) results using batch size 1 with 0-shot evaluation.
>
> [Table R16: Evaluation results on the MBPP+ coding tasks]
> | Method         | Single-pass | +FT | W-bit | Prun. % | R.size (×) | Llama-3.1-8B-Instruct |
> |----------------|:-----------:|:---:|:-----:|:-------:|:-----------:|:-------------:|
> | FP16             |      –      |  –  |  16   |   0%    |     0×    | 75.4% |
> | MoDeGPT     |      x      |  x  |   16   |  15%   |  0.15×  | 42.3%   |
> | SVD-LLM       |      x      |  ✓  |   4   |  15%   |    4.7×  | 24.0% |
> | UniQL (Ours) |      ✓      |  ✓  |   4   |   0%   |     4×    | 64.8%  |
> |                       |      ✓      |  ✓  |   4   |  15%   |    4.7×  | 54.2% |
> |                       |      ✓      |  ✓  |   4   |  25%   |    5.3×  |  33.8%  |
>
>
> > Lack of comparison with unstructured methods: No comparison with popular unstructured pruning methods (e.g., SparseGPT) or hybrid sparse methods is provided.
>
> Unstructured and semi-structured pruning methods require specialized hardware, and therefore fall outside the scope of this work. As a result, we only discuss them in the related work section. Since our method focuses on systematically combining structured pruning and quantization, our main experimental comparisons are made against state-of-the-art methods [4, 5, 8, 16] in this setting.  To address your question, we provide a comparison with semi-structured pruning in Table R17.
>
> [Table R17: Perplexity comparison with structured and semi-structured pruning]
> | Method               | Structure            | 25% | 40% | 50%   |
> |------------------------|-----------------------|-----|-----|--------|
> | SparseGPT (2:4) | Semi-structured |  -   |  -   | 8.15 |
> |SliceGPT              | Structured         | 7.56  | 12.80 | 21.08 |
> | UniQL                  | Structured         | 7.68  |10.83 | 17.01 |

---

> ### Author Response · Authors · 2025-11-20
> **Response to Reviewer 8Vsv (Part 3)**
>
> > Comparison with Unstructured Pruning: How does UniQL compare with unstructured pruning methods like SparseGPT, especially in terms of accuracy-efficiency trade-offs and hardware friendliness?
>
> Our method focuses on systematically combining structured pruning and quantization. Therefore, our main experimental comparisons are made against state-of-the-art methods in weight-only quantization and structured pruning within this setting to ensure a fair evaluation of the accuracy-efficiency trade-offs.
>
> To address the reviewer’s question, we report our perplexity results along with the latency profiling reported in SliceGPT’s ICLR 2024 rebuttal [7] in Table R18. As it is not currently straightforward to run end-to-end evaluations of sparse LLMs,  they provide a micro-benchmark on Llama-2-7B using sparse matrix multiplications implemented with the CuSparseLT 0.5 library on an A100 GPU.
>
> [Table R18: perplexity and speedup comparison with structured and semi-structured pruning]
> | Model    | Method   | WikiText2 PPL | Down Proj (ms) | Up/Gate Proj (ms) | K,V (ms) | Q (ms) | Out (ms) | Total Time (ms) | Relative Speedup |
> |-------------|-------------------|---------------|-----------------|--------------------|----------|---------|----------|------------------|-------------------|
> | Llama-2-7B   | Dense                   | 5.47    | 0.89 | 0.87  | 0.34  | 0.34 | 0.34 | 3.99    | -           |
> | Llama-2-7B   | SparseGPT 2:4     | 8.15    | 0.56 | 0.61  | 0.23  | 0.23 | 0.23 | 2.70    | 1.48×   |
> | Llama-2-7B  | SliceGPT (25%)    | 7.56  | 0.67 | 0.64  | 0.26  | 0.25 | 0.27 | 2.99    | 1.33×   |
> | Llama-2-7B  | SliceGPT (50%)    | 17.17  | 0.46 | 0.44  | 0.18  | 0.18 | 0.18 | 2.06    | 1.94×   |
>
> To complete our comparison, we show that our method outperforms SliceGPT in time-to-last-token (1K+1K) latency in the FP16 precision, as shown in Table R14.
>
> [Table R14: Comparison latency with SliceGPT on A6000]
> | Method   | Prun. Rate % | Bit-width | One-pass | 1K+1K (ms.) |
> |--------------|--------------|----------:|:--------:|:-----:|
> | SliceGPT | 25%          |       16  |    x     | 24202.52 |
> |                 | 40%          |       16  |    x     |  21869.59   |
> |                 | 50%          |       16  |    x     |  18403.68   |
> | UniQL (Ours) | 25%   |       16  |    v     |   19782.73   |
> |                 | 40%          |       16  |    v     |   16256.34  |
> |                 | 50%          |       16  |    v     |   14242.88  |
>
>
> In summary, we can confirm that UniQL achieves higher speedup than SparseGPT on GPU (~1.6× vs. 1.48×) while maintaining comparable perplexity (7.68 vs. 8.15) under the FP16 setting. More importantly, UniQL supports arbitrary low bit-widths and all pruning ratios within a single unified pipeline, enabling broad generalization across diverse hardware platforms without relying on specialized sparsity support. This makes UniQL substantially more flexible and deployment-friendly than methods tied to specific sparse acceleration libraries.

---

### Official Review · Reviewer_oUGV · 2025-10-31

**Soundness:** 2
**Presentation:** 2
**Contribution:** 1
**Rating:** 2
**Confidence:** 4

**Summary:**

This paper proposes a unified quantization and on-device structural pruning method for edge LLMs, including Transformers, State Space Models (SSMs), and hybrid models. To enable the on-device structural pruning, weight sorting is designed for different model blocks, and a LoRA-based recovery fine-tuning (FT) is conducted on the sorted model. 4-bit quantization and pruning with different rates are applied to the model. SVD decomposition is used to reduce quantization error. The evaluation presents different pruning rates that offer a memory reduction of 4x–5.7×.

**Strengths:**

1. The adaptive LLM memory problem discussed in this paper is important and interesting.
2. The proposed methods are evaluated on different model structures.

**Weaknesses:**

1. The contribution is limited, and the proposed methods are very incremental.
a) Methods applied to different model structures appear more as a systematic engineering effort than a novel algorithmic advancement.
b) The quantization, pruning combination has been explored.

2. Insufficient Empirical Evaluation.
a) The paper claims adaptive deployment of LLMs on the edge, but there is no real deployment with different workloads. Crucially, the paper does not address the core systems challenge: how the memory footprint can be dynamically managed at runtime without first loading the entire unpruned model.
b) Unignored performance drop. The reported performance drop is substantial, especially at lower pruning rates (e.g., a 6% drop for Llama2-7B at only 35% pruning).
c) Lack of comparison with related works: Quantization or pruning methods for SSM and hybrid models. Structure pruning methods, such as SliceGPT [1] .

3. The paper writing should be improved.
a) Explain why you do it before the detailed introduction. For example, the reason for weight sorting should be given in the introduction.
b) Too many mathematical symbols make the methods part hard to follow. The algorithms in the Appendix are better for understanding.
c) All of the components should be included in the overview Figure.

[1] Saleh Ashkboos, Maximilian L. Croci, Marcelo Gennari Do Nascimento, Torsten Hoefler, James Hensman. SliceGPT: Compress Large Language Models by Deleting Rows and Columns. ICLR 2024

**Questions:**

1. The finetune is applied after the weight sorter. Why does it apply after quantization? Have you experimented with different orders of the applied methods?
2. The pruning method is similar to SliceGPT [1]. What’s the key difference between SliceGPT and the proposed sorting method?

[1] Saleh Ashkboos, Maximilian L. Croci, Marcelo Gennari Do Nascimento, Torsten Hoefler, James Hensman. SliceGPT: Compress Large Language Models by Deleting Rows and Columns. ICLR 2024

---

> ### Author Response · Authors · 2025-11-20
> **Response to Reviewer oUGV (Part 1)**
>
> We thank the reviewer for their time and effort in evaluating our paper. Below, we highlight our main contributions and provide detailed responses to each question and comment. We look forward to a constructive discussion with the reviewer and to further clarifying and justifying the assigned ratings.
>
> > The contribution is limited, and the proposed methods are very incremental. a) Methods applied to different model structures appear more as a systematic engineering effort than a novel algorithmic advancement.
>
> While we remain open to constructive discussion with the reviewer, we highlight our main contributions below which do not support reviewer’s comment of lack of novelty
>
> To the best of our knowledge, our work is the first to: (1) address flexible model compression under a **novel unified-memory setting** for edge devices, (2) systematically combine quantization and structured pruning with **several new algorithms**, and (3) provide a unified framework applicable to a **wide spectrum of architectures**.
>
> We present a novel compression framework that enables **flexible low-bitwidth models** for edge devices with **unified memory**, applicable to Transformers, State Space Models (SSMs), and hybrid architectures. We introduce several new algorithmic components: a pseudo-inverse-free, quantization-aware SVD, state-aware weight sorting, and a fused rotary embedding kernel, all specifically designed for the proposed unified framework.
>
> In summary, from the problem formulation to the algorithmic and framework design, our work is **far from incremental** or **purely engineering-driven**. To further clarify our contributions, we present a comparison in Table R7 to help reviewers better understand our contributions below.
>
> We would appreciate it if the reviewer could point to concrete recent work that supports their claim so we can engage in a constructive discussion leading to the further improvement of our work.
>
> > b) The quantization, pruning combination has been explored.
>
> We would welcome any related work the reviewer could share in support of this comment. As shown in Figure 4 and Table 9, combining quantization and pruning is non-trivial, which motivates our proposed quantization-aware SVD to address this challenge. In addition, we introduce a new framework featuring novel weight-sorting and masked fine-tuning strategies that enable flexible low-bitwidth models across Transformers, State Space Models (SSMs), and hybrid architectures.
>
> Our goal is to present a novel compression framework that generates flexible low-bitwidth models for edge devices with unified memory. In contrast, prior work [4, 5, 6] does not support adaptive on-device pruning for unified memory on edge and only focuses on fixed-size cloud-based pruning methods for Transformers.
>
> To clarify our contributions, we present a comparison in Table R7 to help reviewers better understand our contributions.
>
> [Table R7: UniQL compared to prior pruning methods]
>
> | Method            | Quant. | Structured Prun. | Joint Design (Quant.+Prun.) | Single-pass | On-device Prun. | Prune KV-cache | Add. Overheads                                  | Supported Arch.                 |
> |-------------------|:------:|:----------------:|:----------------------------:|:-----------:|:----------------:|:--------------:|--------------------------------------------------|----------------------------------|
> | **UniQL (Ours)**  | ✔️ Yes | ✔️ Yes           | ✔️ Yes                       | ✔️ Yes      | ✔️ Yes (single-pass ft)           | ✔️ Yes         | ✘ No                                            | Transformers / SSMs / Hybrid    |
> | SVD-LLM           | ✔️ Yes | ✔️ Yes           | ✘ No                         | ✘ No        | ✘ No (needs ft)             | ✘ No           | ✔️ Yes (U–V computation)                        | Transformers only                |
> | MoDeGPT           | ✘ No   | ✔️ Yes           | ✘ No                         | ✘ No        | ✘ No (needs pinv)            | ✔️ Yes         | ✔️ Yes (extra indexing in RoPE)                 | Transformers only                |
> | SliceGPT          | ✘ No   | ✔️ Yes           | ✘ No                         | ✔️ Yes (but not explicit)     | ✔️ Yes (but not explicit)          | ✘ No           | ✔️ Yes (extra matmul in residual connections)   | Transformers only                |
>
> We believe our work fills a major gap in the current literature by bridging **(1) structured pruning and quantization**, **(2) Transformers and State Space Models**, and **(3) enabling adaptive deployment on edge devices** through a unified-memory architecture.

---

> > ### Author Response · Authors · 2025-11-20
> > **Response to Reviewer oUGV (Part 2)**
> >
> > > Insufficient Empirical Evaluation. a) The paper claims adaptive deployment of LLMs on the edge, but there is no real deployment with different workloads.
> >
> > Due to the page limitations, we provide the most common latency evaluation protocol for edge devices similar to related work [7, 8, 9]. We provide more time-to-last-token latency for more workload in Table R8, R9, and R10 in addition to Table 7 and Table 8 in our manuscript. We will open-source our implementation to facilitate future research on various workloads and settings.
> >
> > [Table R8: Latency evaluated at batch size = 1 with 512 prefill and 512 decode tokens in millisecond (ms)]
> > | Method          | W-bit | Prun. p% | Llama-3.1-8B | Qwen-2.5-7B | Nemotron-H-8B | Mamba2-8B |
> > |-------------------|:------:|:--------:|:-------------:|:------------:|:--------------:|:----------:|
> > | FP16              | 16   |   0%    |   OOM     |   OOM    |    OOM     |  OOM   |
> > | TAO-HQQ      | 4     |   0%   | 76704.32 | 80770.2   |    -    |   -    |
> > | UniQL (Ours) | 4     |   0%   | 42824.66 | 39795.35 |  44419.97  | 41116.36 |
> > |                       | 4     |   35% |  31560.58 | 28185.87 |  30707.68 | 28508.15 |
> >
> > [Table R9: Latency evaluated at batch size = 1 with 128 prefill and 512 decode tokens in millisecond (ms)]
> > | Method          | W-bit | Prun. p% | Llama-3.1-8B | Qwen-2.5-7B | Nemotron-H-8B | Mamba2-8B |
> > |-------------------|:------:|:--------:|:-------------:|:------------:|:--------------:|:----------:|
> > | FP16              | 16   |   0%    |   OOM    |   OOM    |    OOM     |  OOM   |
> > | TAO-HQQ      | 4     |   0%   |  66025.0   |  67199.5  |     -      |    -      |
> > | UniQL (Ours) | 4     |   0%   | 39921.3  |  37723.5   | 29959.0 |  43465.3  |
> > |                       | 4     |   35% |   28120.5  |  26714.9 |  26714.9  |  27828.2   |
> >
> > [Table R10: Latency evaluated at batch size = 1 with 512 prefill and 128 decode tokens in millisecond (ms)]
> > | Method          | W-bit | Prun. p% | Llama-3.1-8B | Qwen-2.5-7B | Nemotron-H-8B | Mamba2-8B |
> > |-------------------|:------:|:--------:|:-------------:|:------------:|:--------------:|:----------:|
> > | FP16              | 16   |   0%    |   OOM    |   OOM    |    OOM     |  OOM   |
> > | TAO-HQQ      | 4     |   0%   |  28083.7  | 29923.8   |    -     |    -    |
> > | UniQL (Ours) | 4     |   0%   | 11692.8 | 10828.1  | 11982.7  | 11159.5 |
> > |                       | 4     |   35% |  8190.1  |   7648.9   | 8263.5   | 7797.3 |
> >
> >
> > > the paper does not address the core systems challenge: how the memory footprint can be dynamically managed at runtime without first loading the entire unpruned model.
> >
> > We note that dynamic memory management is outside the scope of this work and is not yet supported by major inference frameworks such as TensorRT-LLM [1] and SGLang [11]. To achieve the best inference latency, we capture a static CUDA graph; therefore, we prune the model size before deployment under the current framework.
> >
> > In practice, we load and prune the model on the CPU. Specifically, we utilize the available main memory and swap space to prune low bit-width models on-device. After pruning, the model is loaded into the remaining main memory shared between the CPU and GPU. Empirically, this entire process takes 2 minutes using the 6-core Arm Cortex-A78AE 64-bit CPU on Nano 8G. We note that several optimization techniques can be applied to this process, such as parallelizing computations across all layers [6] and concurrently pruning the model to a new size on the CPU while it is being deployed on the GPU.
> >
> > Our work focuses on algorithmic framework design and, to the best of our knowledge, is the first to address the unified memory challenge on modern edge devices. Since the dynamic memory management for CUDA graphs is not yet supported by major inference frameworks, we leave further improvements to future work.

---

> ### Author Response · Authors · 2025-11-20
> **Response to Reviewer oUGV (Part 3)**
>
> > b) Unignored performance drop. The reported performance drop is substantial, especially at lower pruning rates (e.g., a 6% drop for Llama2-7B at only 35% pruning).
>
> Given the lower compression time of our algorithm, we note that our compressed models achieve higher accuracy than state-of-the-art methods such as SVD-LLM [4], MoDeGPT [5], and SliceGPT [6]. In Table R11, we compare all state-of-the-art methods for Llama-2-7B at 35% pruning rate to support our claim. We report average accuracy of zero-shot downstream tasks in the table.
>
> [Table R11: Comparing accuracy at 35% pruning rate]
> | Method     | FT | W-bit | Accuracy (%) |
> |----------------|:--:|:-----:|--------------:|
> | FP16          | -  | 16    | 68.8%         |
> | MoDeGPT  | ✗  | 16     | 56.9%        |
> | SVD-LLM  | ✗  | 16     | 45.0%        |
> | UniQL  | ✗  | 16     | 60.0%        |
> | SVD-LLM   | ✓  | 4     | 56.1%        |
> | UniQL    | ✓ | 4     | 61.0%        |
>
>
> > c) Lack of comparison with related works: Quantization or pruning methods for SSM and hybrid models. Structure pruning methods, such as SliceGPT [1].
>
> SliceGPT [6] has been superseded and is consistently outperformed by MoDeGPT [5]. Therefore, we cite the related papers to acknowledge their contributions but omit SliceGPT [6] from our main experiments. To address the reviewer's concern, we include a comparison in Table R12 with all models evaluated in FP16 for a fair comparison. Given the limited rebuttal time frame, we follow the experimental setting used in MoDeGPT [5] and report our results alongside the metrics they provide. As seen in Table R12, UniQL is superior in performance for the same pruning rate.
>
> [Table R12: Comparison accuracy with SliceGPT and other structured pruning methods]
> | Model      | Compress. | Method                | ARC-e | ARC-c | PIQA  | WinoG. | HellaS. | Average |
> |--------------|-----------|------------------------------|-------|-------|-------|--------|---------|---------|
> | Llama-3-8B | 0%   | Dense                        | 77.69 | 53.58 | 80.63 | 72.69  | 79.16   | 72.75   |
> |                    | 25% | ShortGPT-Alpaca       | 38.13 | 31.40 | 60.94 | 54.22  | 31.52   | 43.24   |
> |                    |          | SliceGPT-Alpaca      | 44.44 | 29.27 | 57.56 | 58.48  | 41.08   | 46.17   |
> |                    |          | MoDeGPT-Alpaca    | 67.05 | 41.13 | 75.52 | 69.61 | 66.49 | 63.96 |
> |                    |         | UniQL                        |  70.37  |  46.33  | 74.16  |  71.82   | 70.12 | 66.56   |
> |                    |         | UniQL-ft                     |  76.05  |  50.00 | 76.55 | 72.93   |  73.37  |  69.78  |
>
>
> > The paper writing should be improved. a) Explain why you do it before the detailed introduction. For example, the reason for weight sorting should be given in the introduction.
>
> We appreciate your constructive suggestions. We have added an explanation on the intuition behind the weight sorting in the beginning of Section 3.2. Please see the updated version of our manuscript.
>
> > b) Too many mathematical symbols make the methods part hard to follow. The algorithms in the Appendix are better for understanding.
>
> We have made every effort to include our novel algorithms within the main paper. Nevertheless, due to page limitations, we provide the pseudo-code in the appendix for completeness. We apologize for the complexity of the notation and believe that the included pseudo-code helps clarify our proposed algorithms.
>
> > c) All of the components should be included in the overview Figure.
>
> We thank the reviewer for this suggestion. We present an overview in Figure 1 to convey the high-level idea of our framework in a concise and intuitive manner, while avoiding unnecessary complexity in details and notations. More detailed algorithmic illustrations are provided in Figures 2, 3, and 4, together with their corresponding sections and detail notations. We have made significant efforts to organize the paper so that readers from different backgrounds can easily understand our ideas and algorithms within the page limit.

---

> ### Author Response · Authors · 2025-11-20
> **Response to Reviewer oUGV (Part 4)**
>
> > The finetune is applied after the weight sorter. Why does it apply after quantization? Have you experimented with different orders of the applied methods?
>
> Our objective is to enable adaptive on-device pruning by sorting weights according to their importance scores, allowing the device to prune the least significant columns. Therefore, following this design principle, our algorithm begins with weight sorting, and then masked fine-tuning.
>
> We provide different pipeline combinations in Table R13 to address your concern. In this table, we experiment with Llama-3.1-8B at a 25% pruning rate using two pipeline orders. Since our work focuses on post-training quantization (PTQ), we place the quantization step at the end of the compression for each combination and do not perform quantization-aware fine-tuning.
>
> [Table R13: Ablation study on pipeline orders]
> | Pipeline                                                                | Bit-width | Prun. Rate | Avg. Acc. |
> |-----------------------------------------------------------------|-------------|----------------|--------------|
> | Llama-3.1-8B                                                        | 16         | 0%              |  74.0%  |
> | Masked fine-tuning → Weight sorting → PTQ      | 4          | 25%             |  66.6%   |
> | Weight sorting → Masked fine-tuning → PTQ      | 4          | 25%             |  67.7%  |
>
>
> > The pruning method is similar to SliceGPT [1]. What’s the key difference between SliceGPT and the proposed sorting method?
>
> We address several key limitations of SliceGPT [6] in our work. SliceGPT [6] applies orthogonal transformations to the hidden-state inputs of each Transformer block to modify the model’s dimensionality. To maintain computational invariance within each block, they introduce an additional linear layer into the residual path. However, this design has several limitations: (1) it adds **extra overhead** to the residual connection; (2) it provides **no memory savings in the KV cache** because the intra-block dimensions (e.g., q_proj, k_proj, v_proj) remain unchanged; and (3) it does **not explore how to integrate quantization** with SliceGPT. In contrast, the above limitations are addressed by UniQL. We present a detailed comparison in Table R7. To complete our comparison, we show that our method outperforms SliceGPT in time-to-last-token (1K+1K) latency in the FP16 precision on the A6000 GPU, as shown in Table R14.
>
> [Table R14: Comparison latency with SliceGPT on A6000]
> | Method   | Prun. Rate % | Bit-width  | 1K+1K (ms.) |
> |--------------|:--------------:|:----------:|:-----:|
> | SliceGPT | 25%          |       16   | 24202.52 |
> |                 | 40%          |       16    |  21869.59   |
> |                 | 50%          |       16   |  18403.68   |
> | UniQL (Ours) | 25%   |       16    |   19782.73   |
> |                 | 40%          |       16    |   16256.34  |
> |                 | 50%          |       16  |   14242.88  |

---

> > ### Author Response · Authors · 2025-11-25
> > **Discussion period reminder**
> >
> > Dear Reviewer oUGV,
> >
> > We hope our rebuttal has addressed your questions and concerns. We look forward to a constructive discussion and would appreciate any justification for your current rating before the discussion period closes.

---

### Official Review · Reviewer_Duby · 2025-10-31

**Soundness:** 3
**Presentation:** 2
**Contribution:** 4
**Rating:** 6
**Confidence:** 3

**Summary:**

The paper introduces UniQL, a unified post-training compression framework designed to efficiently deploy large language models (LLMs) on edge devices. It integrates structured pruning and quantization in a single-shot pipeline that supports adaptive on-device compression based on real-time resource availability.

**Strengths:**

- Unified Framework: UniQL supports Transformers, State Space Models (SSMs), and hybrid architectures, addressing a wide range of LLM structures.

- On-device Adaptive Pruning: Enables users to prune the model at inference time based on the current device memory state.

**Weaknesses:**

- While the results are generally strong, some inconsistencies exist in the evaluation setup:
In Tables 1 and 2, the latency results for different models and methods are evaluated on different hardware platforms (Llama-3.1-8B and Nemotron-H-8B on A6000; Qwen-2.5-7B and Mamba2-8B on Nano 8G). Additionally, Table 2 lacks baseline comparisons such as TRT-AWQ for some models. This raises concerns about the consistency and comparability of latency evaluations across models and methods. Can the authors clarify why all models and methods are not evaluated uniformly across both platforms, and whether such comparisons are fair and meaningful under these mixed settings?

- UniQL is compared to SVD-LLM both with and without fine-tuning, which is helpful. However, the comparison with MoDeGPT is conducted only without fine-tuning, despite UniQL including fine-tuning in its best-performing configuration. Furthermore, all comparisons are conducted at only one sparsity level (15%), which limits the ability to assess how robust each method is across different compression regimes (e.g., 25%, 35%).

- In Table 7, UniQL is evaluated under single-pass adaptive pruning across multiple pruning rates and compared only with SVD-LLM. However, MoDeGPT, another key baseline used in Table 5 and throughout the paper, is not included.

**Questions:**

- With masked fine-tuning (FT), UniQL remains faster (6h 59 m) than both MoDeGPT (7h 03 m) and SVD-LLM (15h 57 m). I do not understand why masked fine-tuning would be faster — could the authors clarify the reason behind this behavior?

I am open to discussing this further during the rebuttal and will be happy to increase my score if my concerns are addressed.

---

> ### Author Response · Authors · 2025-11-20
> **Response to Reviewer Duby (Part 1)**
>
> We thank the reviewer for their positive feedback. Detailed responses to the reviewer’s questions and concerns are provided below. We look forward to constructive discussions to further improve our work and justify the rating.
>
> > inconsistencies exist in the evaluation setup: In Tables 1 and 2, the latency results for different models and methods are evaluated on different hardware platforms. Table 2 lacks baseline comparisons such as TRT-AWQ for some models.
>
> The latency profiling results for all experimental models are presented in Figures 5 and 6 in the Appendix of our manuscript. TensorRT-LLM [1] currently does not support Nano 8G; only Orin AGX is supported [2], so we report TorchAO [3] as our baseline on Nano 8G.
>
> While we aim to include latency profiling for all models across all devices in the main text, page limitations require us to show different models to illustrate the generalization capability of our framework. The complete results are summarized in Table R1 and R2. We average latency of twenty runs and report in millisecond (ms).
>
>
> [Table R1: Full latency profiling on A6000 in ms (1k+1k)]
> | Method         | W-bit | Prun. p% | Llama-2-7B | Llama-3.1-8B | Qwen-2.5-7B | Bamba-v2-9B | Nemotron-H-8B | Mamba2-8B |
> |-------------------|:-----:|:--------:|:-----------:|:-------------:|:------------:|:------------:|:-----------:|:----------:|
> | FP16              | 16    | 0%    | 25913.7 |  26653.8  |  24487.4 |  32363.4 | 25889.4 | 24214.9 |
> | TRT-AWQ      | 4     | 0%     |  9992.7   |   10130.4  | 10070.9   |       -        |        -       |     -       |
> | TAO-HQQ      | 4     | 0%     | 12547.7|   11639.5  |  11960.0  |        -         |       -        |       -      |
> | UniQL (Ours) | 4     | 0%     | 10783.3  |   9944.6   |   9148.4  | 11813.6   | 9095.7  |  8735.1  |
> |                        | 4     | 35%   |  8158.2  |   8105.4   |   7415.3   |  9137.4    |  6955.6  | 6784.5  |
>
> [Table R2: Full latency on Nano 8G in ms (256+256)]
> | Method         | W-bit | Prun. p% | Llama-2-7B | Llama-3.1-8B | Qwen-2.5-7B | Bamba-v2-9B | Nemotron-H-8B | Mamba2-8B |
> |-------------------|:-----:|:--------:|:-----------:|:-------------:|:------------:|:------------:|:-----------:|:----------:|
> | FP16              | 16    | 0%    |   OOM   |     OOM    |    OOM    |   OOM     |    OOM   | OOM   |
> | TAO-HQQ      | 4     | 0%     | 30976.7 |  37717.3  |  38567.9 |       -         |       -        |       -      |
> | UniQL (Ours) | 4     | 0%     | 20435.5 |  20693.0  |  19447.0 |  30109.9  | 22660.5 | 21139.1 |
> |                        | 4     | 35%  |  14319.3  | 14642.4  |  13792.0 |  21048.5  | 15806.7 |14892.5 |

---

> ### Author Response · Authors · 2025-11-20
> **Response to Reviewer Duby (Part 2)**
>
> > the comparison with MoDeGPT is conducted only without fine-tuning, despite UniQL including fine-tuning in its best-performing configuration.
>
> We compare SVD-LLM [4] and MoDeGPT [5] by following their *respective settings* in their papers. To address reviewers’ questions, Table R3 reports the average results **without** fine-tuning, where UniQL applies weight-sorting only and all models are evaluated in FP16. We note that our **single-pass** framework compresses the models once for all compression rates.
>
> [Table R3: Full evaluation results **without** finetuning]
> | Method | Single-pass | +FT | W-bit | Prun. % | R.size (×) | Llama-2-7B | Llama-3.1-8B | Qwen-2.5-7B | Bamba-v2-9B | Nemotron-H-8B | Mamba2-8B |
> |:-----------------|:--:|:--:|:---:|:-----:|:------:|:--------:|:-------:|:------:|:-------------:|:-----------:|:--------:|
> | FP16             | - | - | 16 | 0%   | 0× | 68.8% | 74.0% | 72.4% | 74.6% | 76.0% | 70.6% |
> | MoDeGPT     |x | - |  16 | 15%| 0.15× | 66.2% | 69.4% | 52.1% | – | – | – |
> |                       |x | - |  16 | 25%| 0.25× | 63.4% | 64.9% | 40.8% | – | – | – |
> | SVD-LLM      |x | - |  16 | 15%| 0.15× | 56.3% |  56.7% | 62.6%  | – | – | – |
> |                       |x | - |  16 | 25%| 0.25× | 50.8%   |  45.8%  |  53.2% | – | – | – |
> | UniQL (Ours) |✓| - |  16 | 0% |    0×    | 68.8% |  74.0%  |  72.3% |  74.6%  | 76.0%   |  70.6%  |
> |                       |✓| - |  16 | 15%| 0.15× | 66.7%  |  70.5% |  69.1% | 70.9%  | 68.9% | 65.6% |
> |                       |✓ | - |  16 | 25%| 0.25× | 63.7% |  67.0% |  62.1% | 66.4% | 60.6% | 59.8%|
> |                       |✓ | - |  16 | 35%| 0.35× | 60.0% | 59.0%  |  53.9% |  57.7%  | 48.9% |  52.0%  |
>
>
> For completeness, we also include the **fine-tuned** version of all methods in the Table R4, with all models evaluated in FP16.
>
> [Table R4: Full evaluation results **with** finetuning]
> | Method | Single-pass | +FT | W-bit | Prun. % | R.size (×) | Llama-2-7B | Llama-3.1-8B | Qwen-2.5-7B | Bamba-v2-9B | Nemotron-H-8B | Mamba2-8B |
> |:-----------------|:--:|:--:|:---:|:-----:|:------:|:--------:|:-------:|:------:|:-------------:|:-----------:|:--------:|
> | FP16             | - | -  | 16 | 0%  | 0×      | 68.8% | 74.0% | 72.4% | 74.6% | 76.0% | 70.6% |
> | MoDeGPT     |x |✓| 16 | 15%| 0.15× |  67.0%  | 71.5%  |  61.1%  | – | – | – |
> |                       |x |✓| 16 | 25%| 0.25× |  64.4% | 68.5%  |  55.0%  | – | – | – |
> | SVD-LLM      |x|✓| 16 | 15%| 0.15× |  64.7% | 64.5% |  69.5% | – | – | – |
> |                       |x|✓| 16 | 25%| 0.25× | 62.4% | 59.5%  | 66.8% | – | – | – |
> | UniQL (Ours) |✓|✓ | 16 | 0%| 0×      | 69.2%  |  74.2%   |  73.8%  | 75.6% | 75.1% | 70.1% |
> |                       |✓|✓ | 16 | 15%| 0.15× | 67.2%  | 71.9% |  70.0%  | 72.9% | 73.0% | 66.4% |
> |                       |✓|✓| 16 | 25%| 0.25× | 64.9%  | 69.6% |  65.8%  | 69.7% | 67.3% | 62.7% |
> |                       |✓|✓| 16 | 35%| 0.35× | 61.9%  | 64.0%  |  60.2%  | 65.6% | 60.4% | 58.6% |
>
> Table 2 in our manuscript has been updated accordingly.
>
>
> > all comparisons are conducted at only one sparsity level (15%), which limits the ability to assess how robust each method is across different compression regimes (e.g., 25%, 35%).
>
> We provide results at 15%, 25%, and 35% compression ratios in Table 4 (was Table 7) and compare them with state-of-the-art methods under their most competitive 15% compression setting. Owing to page limitations, we present our experiments concisely to maintain clarity. We appreciate the reviewers’ concern and include the full results for all compression ratios in Table R3, R4, and R5. Tables 2 and 4 in our manuscript have also been updated accordingly.

---

> > ### Author Response · Authors · 2025-11-20
> > **Response to Reviewer Duby (Part 3)**
> >
> > > In Table 7, MoDeGPT, another key baseline used in Table 5 and throughout the paper, is not included.
> >
> > Our comparison in Table 4 (was Table 7) focuses on pruning accuracy under the 4-bit setting. Since MoDeGPT [5] treats quantization as **orthogonal** and omits related quantization studies, we excluded this comparison from Table 4 due to page constraints. The full comparison is available in Table 10 of our manuscript. For completeness, we additionally provide it in Table R5 to address the reviewers’ concern. Table 4 in our manuscript has also been updated accordingly.
> >
> > [Table R5: Full evaluation results with finetuning and 4-bit quantization]
> > | Method | Sing-pass | +FT | W-bit | Prun. % | R.size (×) | Llama-2-7B | Llama-3.1-8B | Qwen-2.5-7B | Bamba-v2-9B | Nemotron-H-8B | Mamba2-8B |
> > |:-------------------|:--:|:--:|:---:|:-----:|:------:|:--------:|:-------:|:------:|:-------------:|:-----------:|:--------:|
> > | FP16              | - | -  | 16 | 0%  | 0×      | 68.8% | 74.0% | 72.4% | 74.6% | 76.0% | 70.6% |
> > | MoDeGPT      |x | ✓ |  4 | 15%| 0.15× | 63.7% | 64.2% | 52.2%  | – | – | – |
> > |                        |x | ✓ |  4 | 25%| 0.25× | 60.3% | 59.3%  | 48.4%  | – | – | – |
> > | SVD-LLM       |x | ✓ |  4 | 15% | 4.7×  | 63.2% | 60.6%  | 66.8% | – | – | – |
> > |                       |x | ✓ |  4 | 25% | 4.7×  | 59.1%  |  54.2% | 64.6% | – | – | – |
> > | UniQL (Ours) |✓ | ✓ |  4 | 0%   | 4×    | 67.6% | 73.6%  | 72.4% | 75.1% | 73.3% | 69.3% |
> > |                       |✓ | ✓ |  4 | 15% | 4.7× | 65.6% | 71.4%  | 68.1% | 70.3% | 70.5% | 65.8% |
> > |                       |✓ | ✓ |  4 | 25% | 5.3× | 63.5% | 67.7%  | 64.0% | 67.4% | 64.7% | 61.8% |
> > |                       |✓ | ✓ |  4 | 35% | 6.1× | 61.0% | 62.7%  | 58.1% | 62.7% | 59.0% | 57.7% |
> >
> >
> > >I do not understand why masked fine-tuning would be faster — could the authors clarify the reason behind this behavior?
> >
> > Pseudo-inverse has a complexity of $O(n^3)$ for a $n$-size squared matrix. This is particularly time-consuming when computing the pseudo-inverse of correlation matrices in MLP layers because $D_{\mathrm{int}}$ is a large number in most LLM designs, *e.g.*, Llama-3-8B $D_{\mathrm{int}} = 14336$. Moreover, Pseudo-inverse computation requires a high-precision FP64 to maintain numerical stability, which demands substantial memory usage for full-precision weights. We highlight the line 197-203 in the updated manuscript.
> >
> > We profile the pseudo-inverse latency for FP64 square matrices of size [k, k] on an A6000 GPU and report the latency in minutes (min.) in Table R6. This result is included in Table 1 of our manuscript to better illustrate the underlying idea.
> >
> > [Table R6: Pseudo-inverse latency]
> > | Matrix Size [k, k] | Latency (min.) |
> > |------------------------|-------------------:|
> > | [1024,  1024]       | 0.02 |
> > | [4096,  4096]       | 0.57 |
> > | [8192, 8192]        | 4.24 |
> > | [14336, 14336]    | 20.5  |

---

### Author Response · Authors · 2025-11-20
**Global response**

We thank all reviewers for their time and effort in evaluating our manuscript. We appreciate the thoughtful feedback, which has helped us further refine and strengthen the work, and we have revised the manuscript accordingly. Reviewer comments are highlighted in yellow, and newly added content is marked in blue. Below, we summarize the key contributions of our work.

## Novelty and contributions (Reviewer oUGV and ZDeG)

To the best of our knowledge, our work is the first to: **(1)** address **flexible model compression** under a novel **unified-memory setting** for edge devices, **(2)** systematically combine quantization and structured pruning with **several new algorithms**, and **(3)** provide a unified framework applicable to **most popular model architectures**.

We present a novel compression framework that enables **flexible n-bit (i.e., 3, 4-bit) models** for edge devices with **unified memory**, applicable to **Transformers**, **State Space Models** (SSMs), and **hybrid architectures**. We introduce several new algorithmic components: a pseudo-inverse-free, quantization-aware SVD, state-aware weight sorting, and a fused rotary embedding kernel, all specifically designed for the proposed unified framework.

To clarify our contributions, we present a comparison in **Tables R7** and **R19** to help reviewers better understand our contributions below.

## Baselines, latency and ablation study (Reviewer Duby, oUGV, 8Vsv and ZDeG)
Due to the page limit at submission time, we included only the most representative experiments and strongest baselines to demonstrate our core idea. In response to the reviewers’ feedback, we have added and updated all requested experiments and comparisons, which substantially strengthen the work. All additional experimental tables introduced in the rebuttal are labeled with the prefix “R” (e.g., Table R1, Table R2) as enumerated below, while tables without the “R” prefix refer to those already included in the main manuscript.

## List of table used in the rebuttal
- Table R1: Full latency profiling on A6000 in ms (1k+1k)
- Table R2: Full latency on Nano 8G in ms (256+256)
- Table R3: Full evaluation results without finetuning
- Table R4: Full evaluation results with finetuning
- Table R5: Full evaluation results with finetuning and 4-bit quantization
- Table R6: Pseudo-inverse latency
- Table R7: UniQL compared to prior pruning methods
- Table R8: Latency evaluated at batch size = 1 with 512 prefill and 512 decode tokens in millisecond (ms)
- Table R9: Latency evaluated at batch size = 1 with 128 prefill and 512 decode tokens in millisecond (ms)
- Table R10: Latency evaluated at batch size = 1 with 512 prefill and 128 decode tokens in millisecond (ms)
- Table R11: Comparing accuracy at 35% pruning rate
- Table R12: Comparison accuracy with SliceGPT and other structured pruning methods
- Table R13: Ablation study on pipeline orders
- Table R14: Comparison latency with SliceGPT
- Table R15: Ablation study on calibration sets
- Table R16: Evaluation results on the MBPP+ coding tasks
- Table R17: Perplexity comparison with structured and semi-structured pruning
- Table R18: Perplexity and speedup comparison with structured and semi-structured pruning
- Table R19: UniQL compared to prior quantization methods
- Table R20: Experimental results of 3-bit UniQL
- Table R21: Comparison between weight-only and weight-activation quantization
- Table R22: Generation throughput comparison for Llama-2-7B on A100
- Table R23: Accuracy comparison for Llama-2-7B across quantization methods
- Table R24: Hadamard matrix fusion for Transformer blocks
- Table R25: Hadamard matrix fusion for Mamba blocks

## List of modification of our manuscript
- Add pseudo-inverse latency in Table 1 (reviewer Duby)
- Highlight line 158-159 and add brief intuitions (reviewer oUGV)
- Highlight line 197-203 (reviewer Duby)
- Highlight line 248-249 and 253-254 (reviewer 8Vsv)
- Highlight line 278-281 (reviewer 8Vsv)
- Append 25% pruning rate to Table 2 (reviewer Duby)
- Add “weight-only” in line 416 and Table 3 caption (reviewer ZDeG)
- Add 4-bit MoDeGPT with 15% and 25% pruning rates, and SVD-LLM with 25% pruning rate in Table 4 (reviewer Duby)
- Add “Additional ablation studies can be found at Appendix C.” (reviewer Duby, oUGV, 8Vsv)
- Reorganize our experiment sections for better readability (reviewer oUGV)
- Add B.1 COMPARISON WITH ADDITIONAL BASELINES (reviewer oUGV)
- Add B.3 EVALUATION ON CODING TASKS (reviewer 8Vsv)
- Add C.1 ABLATION STUDY ON CALIBRATION SETS (reviewer 8Vsv)
- Add C.2 ABLATION STUDY ON 3-BIT UNIQL (reviewer ZDeG)
- Add F.3 HADAMARD TRANSFORM FUSION (reviewer ZDeG)
- Add G ENERGY PROFILING

---

> ### Author Response · Authors · 2025-11-20
> **List references used in the rebuttal**
>
> - [1] NVIDIA. TensorRT-LLM: High-performance inference for Large Language Models. https://github.com/NVIDIA/TensorRT-LLM, 2023. Accessed: 2025-09-13.
> - [2] NVIDIA Jetson AI Lab. TensorRT-LLM for Jetson. https://www.jetson-ai-lab.com/tensorrt_llm.html
> - [3] torchao. Torchao: Pytorch-native training-to-serving model optimization, oct 2024. URL https:
> //github.com/pytorch/ao.
> - [4] Wang, Xin, et al. "Svd-llm: Truncation-aware singular value decomposition for large language model compression." ICLR 2025.
> - [5] Lin, Chi-Heng, et al. "Modegpt: Modular decomposition for large language model compression." ICLR 2025.
> - [6] Ashkboos, Saleh, et al. "Slicegpt: Compress large language models by deleting rows and columns." ICLR 2024.
> - [7] Ashkboos, Saleh, et al. "Slicegpt: Compress large language models by deleting rows and columns." ICLR 2024 OpenReview https://openreview.net/forum?id=vXxardq6db&noteId=Ux3B55wumF
> - [8] Lin, Ji, et al. "Awq: Activation-aware weight quantization for on-device llm compression and acceleration." MLSYS 2024.
> - [9] Chiang, Hung-Yueh, et al. "Quamba2: A Robust and Scalable Post-training Quantization Framework for Selective State Space Models." ICML 2025.
> - [10] Austin, Jacob, et al. "Program synthesis with large language models." arXiv preprint arXiv:2108.07732 (2021).
> - [11] Zheng, Lianmin, et al. "Efficiently Programming Large Language Models using SGLang." (2023).
> - [12] Liu, Zechun, et al. "Spinquant: Llm quantization with learned rotations." ICLR 2025.
> - [13] Ashkboos, Saleh, et al. "Quarot: Outlier-free 4-bit inference in rotated llms." NeurIPS 2024
> - [14] Zhao, Yilong, et al. "Atom: Low-bit quantization for efficient and accurate llm serving." MLSYS 2024
> - [15] Lin, Yujun, et al. "Qserve: W4a8kv4 quantization and system co-design for efficient llm serving." MLSYS 2025
> - [16] Badri, Hicham, and Appu Shaji. "Half-quadratic quantization of large machine learning models." (2023).

---

### Author Response · Authors · 2025-12-03
**Rebuttal Summarization for ACs and PCs**

# Strengths from all reviewers
- **Unified, Broad Architectural Coverage**: UniQL supports Transformers, SSMs, and hybrid architectures, offering a unified post-training compression pipeline in one-pass fashion. (Duby, 8Vsv, ZDeG)
- **Novel On-Device Adaptive Pruning**: Enables dynamic pruning at inference time based on device memory or workloads, useful for edge deployment. (Duby, 8Vsv)
- **Practicality for Edge Deployment**: A one-shot, unified pipeline without retraining, specifically targeting edge scenarios.(ZDeG, oUGV)
- **Strong System-Level Engineering**: Includes fused RoPE kernel, quantization-aware SVD (QSVD), SSM-aware sorting, and avoidance of pseudo-inverses. (8Vsv, ZDeG)
- **High Compression Efficiency**: Up to 22× speedup over MoDeGPT by optimizing decomposition and removing pseudo-inverse steps. (8Vsv)
- **Extensive Experimentation Across Models and Hardware**: Demonstrated on Llama, Qwen, Mamba, and multiple hardware platforms (A6000, Orin Nano 8G). (8Vsv)
- **Practical Export Path**: Quantizes embeddings and LM head to 4-bit, reducing total footprint beyond typical W4A16 methods. (ZDeG)

---

> ### Author Response · Authors · 2025-12-03
> **Rebuttal Summarization for ACs and PCs**
>
> # Weaknesses from all reviewers and our rebuttal with additional experiments
>
> ### Novelty & Conceptual Contribution Concerns (oUGV, ZDeG)
> - Our work is the first to: **(1)** address flexible model compression under a novel **unified-memory** setting for edge devices, **(2)** systematically combine quantization and structured pruning with **several new algorithms**, and **(3)** provide a unified framework applicable to most **popular model architectures**.
> - Table R7: UniQL compared to prior pruning methods
> - Table R19: UniQL compared to prior quantization methods
>
> ### Inconsistent Evaluation Hardware & Baselines (Duby, oUGV)
> - We will **open-source** our implementation to facilitate future research on various workloads and settings.
> - Table R1: Full latency profiling on A6000 in ms (1k+1k)
> - Table R2: Full latency on Nano 8G in ms (256+256)
> - Table R8: Latency evaluated at batch size = 1 with 512 prefill and 512 decode tokens in millisecond (ms)
> - Table R9: Latency evaluated at batch size = 1 with 128 prefill and 512 decode tokens in millisecond (ms)
> - Table R10: Latency evaluated at batch size = 1 with 512 prefill and 128 decode tokens in millisecond (ms)
>
> ### Uneven Fine-tuning Comparisons and Limited Sparsity-Rate Exploration (Duby)
> - Table R3: Full evaluation results without finetuning
> - Table R4: Full evaluation results with finetuning
> ### Missing Baselines in Adaptive Pruning Experiments (Duby)
> - Table R5: Full evaluation results with finetuning and 4-bit quantization
>
> ### Insufficient Comparison to Related Work
>
> **SparseGPT (unstructured pruning) (8Vsv)**
> - Our method **outperforms** SliceGPT, SparseGPT, and other structured pruning methods in both accuracy and latency
> - Table R17: Perplexity comparison with structured and  semi-structured pruning
> - Table R18: Speedup comparison with structured and semi-structured pruning
>
> **SliceGPT (structured pruning) (oUGV)**
> - Our method **outperforms** SliceGPT, SparseGPT, and other structured pruning methods in both accuracy and latency
> - Table R7: UniQL compared to prior pruning methods
> - Table R12: Comparison accuracy with SliceGPT and other structured pruning methods
> - Table R14: Comparison latency with SliceGPT
> - Table R17: Perplexity comparison with structured and  semi-structured pruning
> - Table R18: Speedup comparison with structured and semi-structured pruning
>
> **PTQ baselines: AWQ, QServe, Quarot, SpinQuant (ZDeG)**
> - Our weight-only quantization approach delivers **comparable** or **superior** in accuracy and latency to weight-activation PTQ methods and offers improved flexibility for edge-device deployment, although such comparisons are not the primary focus of our work.
> - Table R19: UniQL compared to prior quantization methods
> - Table R21: Comparison between weight-only and weight-activation quantization
> - Table R22: Generation throughput comparison for Llama-2-7B on A100
> - Table R23: Accuracy comparison for Llama-2-7B across quantization methods
>
> **Lack of Comparison: Quantization+Pruning vs Lower-bit Quantization (ZDeG)**
> - The comparison between “quantization + pruning” and “lower-bit quantization alone” is **not** the primary focus of this work. Our goal is to present a novel compression framework that generates **flexible n-bit models** for edge devices with unified memory.
> - Table R20: Experimental results of 3-bit UniQL
>
> ### Accuracy Degradation at High Pruning Rates (8Vsv, oUGV)
> - Given the **lower compression time** of our algorithm, we note that our compressed models achieve **higher accuracy** than state-of-the-art methods such as SVD-LLM [4], MoDeGPT [5], and SliceGPT [6].
> - Table R11: Comparing accuracy at 35% pruning rate
>
> ### Boarder evaluation results (8Vsv)
> - We show the superior performance to baselines on the coding task
> - Table R16: Evaluation results on the MBPP+ coding tasks
>
> ### Pipeline orders and Calibration Sensitivity (oUGV, 8Vsv)
> - Table R13: Ablation study on pipeline orders
> - Table R15: Ablation study on calibration sets
>
> ### Rotation-Pruning Interaction Not Fully Explained (ZDeG)
> - Table R24: Hadamard matrix fusion for Transformer blocks
> - Table R25: Hadamard matrix fusion for Mamba blocks
>
> ### System-Level Demonstrations (oUGV, 8Vsv)
> - In practice, we load and prune the model on the CPU. Specifically, we utilize the available main memory and swap space to prune low bit-width models on-device. After pruning, the model is loaded into the remaining main memory shared between the CPU and GPU. Empirically, this entire process takes 2 minutes using the 6-core Arm Cortex-A78AE 64-bit CPU on Nano 8G. We note that several optimization techniques can be applied to this process, such as parallelizing computations across all layers [6] and concurrently pruning the model to a new size on the CPU while it is being deployed on the GPU.
>
> ### Writing and Presentation Issues (oUGV)
> - We reorganize our experiment sections for better readability in the updated manuscript

---

### Meta-Review · Area_Chair_EY5U · 2026-01-04

**Summary:**

UniQL is a one-shot post-training compression framework that jointly integrates low-bit quantization and structured (low-rank / column) pruning in a single pipeline, and is designed to support Transformers, SSMs (e.g., Mamba), and hybrid models while enabling on-device configurable pruning rates for edge deployment.

The rebuttal has clearly strengthened the empirical analysis (multi-device latency tables, multi-rate pruning sweeps, added baselines, ablations).
The question remains whether the the ICLR community accepts systems integration + unified applicability + practical adaptive pruning as sufficient novelty, and whether the lack of a matched-budget W3 vs (W4+pruning) story remains a decisive weakness (as it probably does for one borderline reviewer).
Considering the practical relevance for the community if the code and quantized models are released, the AC leans towards acceptance of the paper.

**Reviewer Concerns:**

**Reviewer Duby.**

The concerns were mostly addressed:
* Mixed hardware / missing latency clarity: fixed via R1/R2 + framework support explanation.
* Fairness of FT comparisons + sparsity exploration: added R3/R4 with multi-rate results and manuscript updates.
* Missing MoDeGPT in pruning+quant tables: added R5 and updated tables.
* Masked FT speed confusion: pseudo-inverse latency explanation + R6.


**Reviewer oUGV.**

Addressed concerns:
* Pipeline order question: ablation provided (R13).
* SliceGPT similarity/difference: detailed explanation + accuracy (R12) + latency (R14) + positioning (R7).
* More workload-style latency reporting: R8–R10.
* Writing/structure: claimed reorganizations + added intuition + added appendix sections.

Still outstanding:
* Core systems challenge (true dynamic runtime memory management without loading the full model): largely not solved, declared out-of-scope.
* Novelty framing may remain unconvinced despite R7 and novelty argumentation (this was a central basis for their “Contribution: 1”).
* Accuracy drop concerns may persist (even if comparatively better than baselines).


**Reviewer 8Vsv.**

Addressed concerns:
* Calibration sensitivity: ablation (R15) + broader eval references (MMLU) + coding task (R16).
* Unstructured/semi-structured comparison: added SparseGPT comparisons (R17/R18).
* Interpretability: added figures/algorithms pointers for QSVD and state-aware parts.
* Accuracy at high pruning: at least one clear comparative table at 35% (R11).

Still outstanding:
* Device-side pruning overhead: only a coarse “~2 minutes CPU prune” statement; no detailed overhead breakdown.
* High-pruning accuracy for the specifically-cited weak case (e.g., Mamba2) may not fully be resolved.


**Reviewer ZDeG.**

Addressed concerns:
* Rotation/pruning mismatch clarity: explicitly answered with R24/R25.
* PTQ baseline strength: partially improved with R22/R23 and clearer scope positioning (R19/R21).
* Novelty positioning: strengthened via explicit comparative tables and claimed “first unified-memory + unified arch support + joint design.”

Still outstanding:
* The key matched-budget argument: “W4+pruning vs W3 alone” remains not convincingly answered, since the rebuttal declines the requested Pareto/matched comparison and provides only an internal 3-bit UniQL table (R20) rather than the comparative evidence the reviewer asked for. \

**Reviewer Scores:**

Reviewer Duby
They likely would have kept their score 6 (possibly increased it to 8, as they indicated that they were open to a score increase).

Reviewer oUGV
The rebuttal strengthens comparisons and clarity, but it does not directly satisfy the reviewer’s “core challenge” framing or their novelty bar.
Therefore, they would have likely increased their score maximally to 4.

Reviewer 8Vsv
They likely would have kept their score 8.

Reviewer ZDeG
They might have increased their score to 6 if they accepted the scope argument and are satisfied by the added baselines + rotation clarification.

---

### Decision · Program_Chairs · 2026-01-26

Accept (Poster)